# Deep learning-enabled multi-organ segmentation in whole-body mouse scans

Oliver Schoppe [1,2,3✉], Chenchen Pan[3,4], Javier Coronel[1,2], Hongcheng Mai[3,4], Zhouyi Rong [3,4], Mihail Ivilinov Todorov[3,4,5], Annemarie Müskes[6], Fernando Navarro[1,2], Hongwei Li[1], Ali Ertürk [3,4,7✉] & Bjoern H. Menze [1,2,8,9✉]

Whole-body imaging of mice is a key source of information for research. Organ segmentation is a prerequisite for quantitative analysis but is a tedious and error-prone task if done manually. Here, we present a deep learning solution called AIMOS that automatically segments major organs (brain, lungs, heart, liver, kidneys, spleen, bladder, stomach, intestine) and the skeleton in less than a second, orders of magnitude faster than prior algorithms. AIMOS matches or exceeds the segmentation quality of state-of-the-art approaches and of human experts. We exemplify direct applicability for biomedical research for localizing cancer metastases. Furthermore, we show that expert annotations are subject to human error and bias. As a consequence, we show that at least two independently created annotations are needed to assess model performance. Importantly, AIMOS addresses the issue of human bias by identifying the regions where humans are most likely to disagree, and thereby localizes and quantifies this uncertainty for improved downstream analysis. In summary, AIMOS is a powerful open-source tool to increase scalability, reduce bias, and foster reproducibility in many areas of biomedical research.

[1] Department of Informatics, Technical University of Munich, Munich, Germany. [2] Center for Translational Cancer Research (TranslaTUM), Klinikum rechts der Isar, Technical University of Munich, Munich, Germany. [3] Institute for Tissue Engineering and Regenerative Medicine (iTERM), Helmholtz Zentrum München, Neuherberg, Germany. [4] Institute for Stroke and Dementia Research (ISD), University Hospital, LMU Munich, Germany. [5] Graduate School of Systemic Neurosciences (GSN), Munich, Germany. [6] Berlin-Brandenburg Center for Regenerative Therapies, Charité, Universitätsmedizin Berlin, Berlin, Germany. [7] Munich Cluster for Systems Neurology (SyNergy), Munich, Germany. [8] Institute for Advanced Study, Department of Informatics, Technical University of Munich, Munich, Germany. [9] Department of Quantitative Biomedicine, University of Zurich, Zurich, Switzerland. ✉email: oliver.schoppe@tum.de; erturk@helmholtz-muenchen.de; bjoern.menze@tum.de

Animal models are the backbone of many areas of biomedical and preclinical research and the mouse is the most commonly used organism to study the diseases that occur in humans[1,2]. Understanding and characterizing mouse models in detail is considered as a key to improving the reproducibility for human applications[3]. Due to its fast technological advancement, whole-body imaging with diverse modalities plays an increasingly important role in murine research[4–14]. For a broad range of research areas, from cancer[15–17] to organ lesion studies [18], from radiation studies[19] to drug delivery[20,21] and nanoparticle uptake[22–27], quantitative and comparative analyses of the acquired imaging data require segmentation of the mouse anatomy. Delineation of the major organs and other structures of interest allows the extraction of quantitative information such as organ shape and size, drug uptake, signals from biomarkers, or metastatic distribution from the imaging data.

Thus, volumetric organ segmentation is an essential step in data analysis for many areas of biomedical research. Traditionally, this task is performed manually by delineating the organ outlines with polygons in each slice of the volumetric scan. However, while this procedure requires great attention to detail and expertize in anatomy and the imaging modality, it is highly repetitive and time-consuming. This limits the sample size that can be analyzed with justifiable efforts. To reduce the time needed for manual segmentation, interactive, and semiautomatic procedures (such as thresholding and region-growing) can be used[28] but may go along with imprecise delineations. Furthermore, manual segmentation is a difficult task even for experts and tends to be prone to human error and bias, especially in low-contrast modalities like CT, which can negatively affect the reproducibility and objectivity of the obtained results. Thus, there is a great need for automatic organ segmentation, which has been an active field of research for decades[29–32]. The state-of-the-art and most commonly used approaches for segmentation of mouse organs in volumetric scans make use of one or several anatomical reference atlases, which are mapped to the scan by means of (elastic) deformation and thus also make prior assumptions on shape and size[33–37]. Other approaches to mouse organ segmentation include learning-based techniques such as support-vector machines and random forests[38,39]. Unfortunately, these automatic approaches often do not reach satisfactory segmentation quality and still take several minutes to process a single scan. The limitations of the approaches so far are especially apparent for organs with low imaging contrast or complex or variable shapes (for example, the bladder[33], the stomach[38], or the spleen[36,40]); for the same reasons, these organs are difficult to segment accurately even for human experts (as we show). But the general rise of deep neural networks in biomedical image processing suggests that learnable convolutional kernels provide the most promising basis for automatic organ segmentation in human and in murine data[40–43]. However, so far, there was no single end-to-end deep neural network that directly segments a large number of organs in mouse scans at high quality.

Here, we present a fully integrated pipeline based on a single end-to-end deep neural network for organ segmentation of volumetric whole-body scans of mice termed AIMOS (AI-based Mouse Organ Segmentation). Without any human intervention or parameter-tuning, it can segment the main organs (brain, lungs, heart, liver, kidneys, spleen, bladder, stomach, and intestine) and the skeleton in volumetric scans of mice. Here, AIMOS was trained and tested on four different whole-body imaging modalities and variants: native micro-CT[28], contrast-enhanced micro-CT[28], native light-sheet microscopy, and nucleus-staining fluorescence microscopy (using propidium iodide (PI) as the staining agent). We validated the performance on over 220 annotated whole-body scans of mice, the largest reported validation yet to the best of our knowledge. With a processing time for a whole-body scan of less than one second, AIMOS works two orders of magnitude faster than the commonly used atlas-registration-based techniques. Furthermore, the segmentation quality matches or exceeds those of state-of-the-art approaches and, importantly, matches the quality of manual segmentation by human experts, even for low-contrast organs such as the spleen in CT. We show that the pretrained models from this study allow fast retraining of AIMOS on other modalities with only a few annotated scans. The code, the trained algorithm, and the annotated datasets for AIMOS are open source and freely available to the scientific community.

So far, the performance of automated methods has been typically assessed by quantifying the volumetric overlap between the predicted segmentation and a human expert annotation through the SrensenDice score. The human annotation is taken as an absolute reference (often called ground truth)[44]. However, it is well-known that manually created reference segmentations of medical images are affected by subjective interpretation and thus suffer from errors and individual bias[45,46]. Diverging human annotations have previously been addressed, for example by training models to predict a set of diverse but plausible segmentations[47,48]. However, solely comparing a single model prediction to a single human reference annotation, a common practice in the field[34–40], fails to appreciate that the reference a) may be suboptimal and b) reflects the subjective interpretation of an individual. To address this issue, we analyzed this shortcoming in detail and showed that a predicted segmentation with a higher Dice score does not necessarily reflect a better solution, but may just be a sign of overfitting to the subjective interpretation of the reference annotator. While a model can be trained to closely follow the interpretation of a given annotator, an independent expert may have a different interpretation of the same data and would disagree with the proposed solution. Thus, we show that at least two independently created reference annotations are required to assess the true model performance. Inspired by this observation, we enhanced AIMOS to a dual approach: predicting organ segmentations with concurrent identification of ambiguous image regions where human experts are most likely to disagree in their interpretation. Altogether, AIMOS not only provides highly precise delineations of mouse organs but also quantifies and localizes uncertainty, an essential step for further statistical analysis.

In summary, this study makes several contributions. We provide a deep learning-based processing pipeline termed AIMOS, the fastest and most accurate solution for the automated segmentation of the major organs and the skeleton in volumetric mouse scans. The trained models were tested in the largest validation yet on a dataset with over 220 manually annotated mouse scans. AIMOS can be directly transferred to other modalities or animal models. The code, the trained models, and the annotated data are freely available to the public, fostering reproducibility and further improvement. Furthermore, AIMOS identifies ambiguous image regions where human annotators are most likely to disagree in their interpretations. Together, AIMOS provides researchers with a versatile tool that enables fast, unbiased, and interpretable quantifications of mouse scans and thus can dramatically increase scalability and reproducibility in imaging-based murine research. Areas of application include, but are not limited to, the spread of cancer metastases, organ lesion studies, radiation studies, drug delivery and nanoparticle uptake assessment, analysis of pathogen infections, as well as general localized studies of organ-related pathologies.

## Results

**AIMOS pipeline**. We developed AIMOS, a fully automatic deep learning-enabled data processing pipeline for the segmentation of volumetric whole-body scans of mice (Fig. 1). While the general approach is not tied to a specific type of input and can

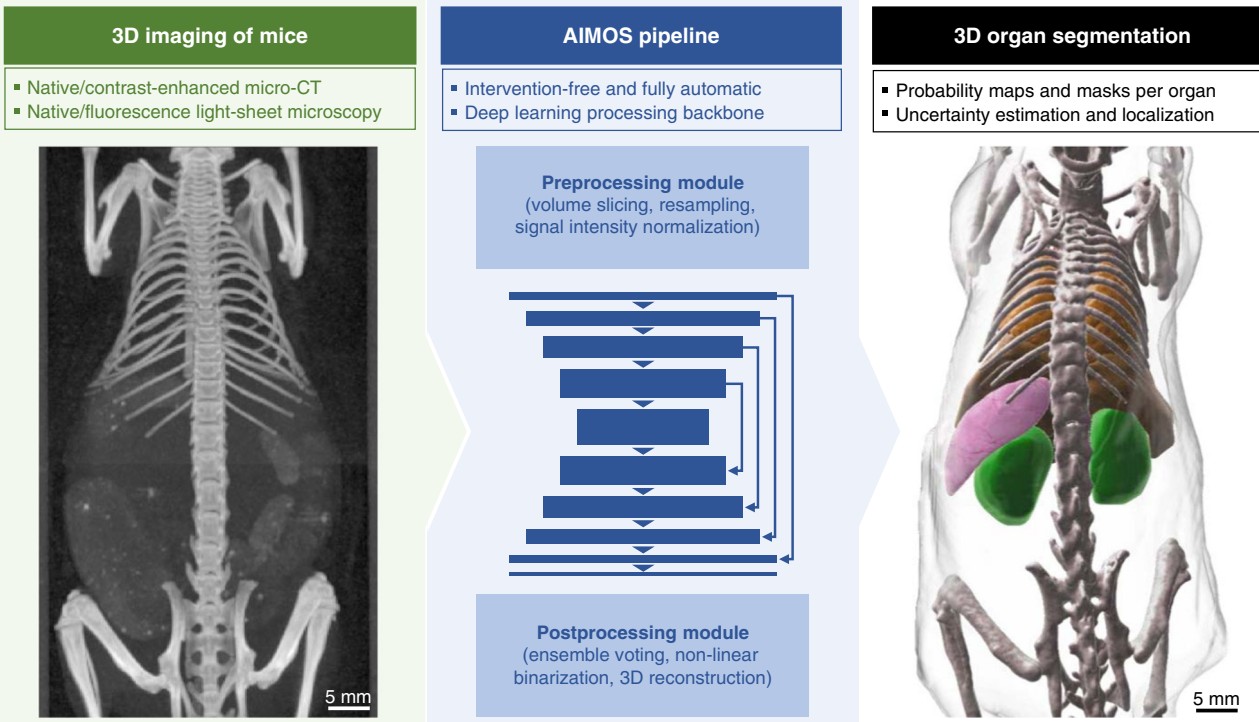

**Fig. 1 The AIMOS processing pipeline.** AIMOS is a fully automatic, end-to-end deep learning pipeline that segments multiple organs of mice in 3D scans across different imaging modalities.

also be transferred to other imaging modalities, here we trained and evaluated the AIMOS pipeline on four different modalities and variants: native micro-CT, contrast-enhanced micro-CT, native light-sheet microscopy, and nucleus-staining fluorescence light-sheet microscopy. The pipeline takes a volumetric scan as its input and predicts 3D delineations of the main organs. The pipeline consists of a preprocessing module, a deep learning backbone, and a postprocessing module. The preprocessing module slices the volume into 2D images, normalizes the signal intensity, and resamples them to a desired resolution. While we processed the scans at a resolution of 240 μm/*vx*, we performed an ablation study found that AIMOS works well over a fairly broad range of resolutions from 120 μm/*vx* to 1120 μm/*vx* (Supplementary Fig. 1a). As the backbone, we chose a U-Net-like convolutional neural network with six levels of encoding and decoding blocks with up to 1024 feature channels. Again, we performed an ablation study to assess the sensitivity of the pipeline to model complexity (number of levels), and found that leaner models performed equally well (Supplementary Fig. 1b). The neural network creates a probability map for each organ. The postprocessing module comprises binarization and volumetric reconstruction of the output. As an optional feature, it also comprises ensemble-voting, in which the predictions of a number of independently trained networks are merged into one ensemble prediction (see Methods section for details). Ensemble-voting is known to improve biomedical image segmentations by reducing variance from statistical fluctuations[49,50]. The effect of ensemble-voting is further analyzed below.

**AIMOS segments organs with human accuracy.** For the first set of experiments, we trained and evaluated AIMOS on two publicly available datasets, native microCT and contrast-enhanced microCT whole-body scans, that were explicitly designed to foster the development of deep learning-based organ segmentations[28].

With over 220 manually annotated scans, this represents the largest validation, to the best of our knowledge, of an automatic mouse organ segmentation solution reported so far. The datasets and any preprocessing is described in the Methods and a comprehensive description of the datasets is available in the accompanying publication[28]. Supplementary Fig. 2 shows the structure and representative samples of both datasets.

To evaluate the performance of the AIMOS processing pipeline, we trained and tested it on both CT datasets. We quantified the overlap between predicted organ segmentations and the reference annotation from a human expert. Additionally, we then compared processing speed and performance of AIMOS with other common approaches. Furthermore, we also performed a qualitative assessment of the physiological plausibility of the predicted organ segmentations, and their deviations from the reference annotations. Importantly, while this evaluation was carried out on the entire dataset, the test predictions were exclusively made for mice that were not used during training (k-fold cross-validation; see Methods section for details).

For the native microCT dataset, the median Dice scores varied between 88 and 93% for heart, lungs, liver, bladder, intestines, and both kidneys (Fig. 2a). Interquartile ranges varied from 80 to 95%, suggesting that the organs are consistently segmented with high precision. Importantly, the median Dice scores are all above the Dice scores of two independent human expert annotations (blue lines), indicating that AIMOS matches or exceeds human performance in organ segmentation. This even holds for the spleen (by an even bigger margin of over 30 p.p.), although the overall performance and variance are not as good (median Dice score of 73%). This may be largely explained by the low CT-contrast for this organ.

For the contrast-enhanced micro-CT dataset (Fig. 2b), the contrast of the spleen was boosted by injecting a contrast agent. This enables humans to segment the spleen much more consistently (82% vs. 37%). Also, AIMOS reached a substantially higher Dice score for the spleen (89%), matching the already high

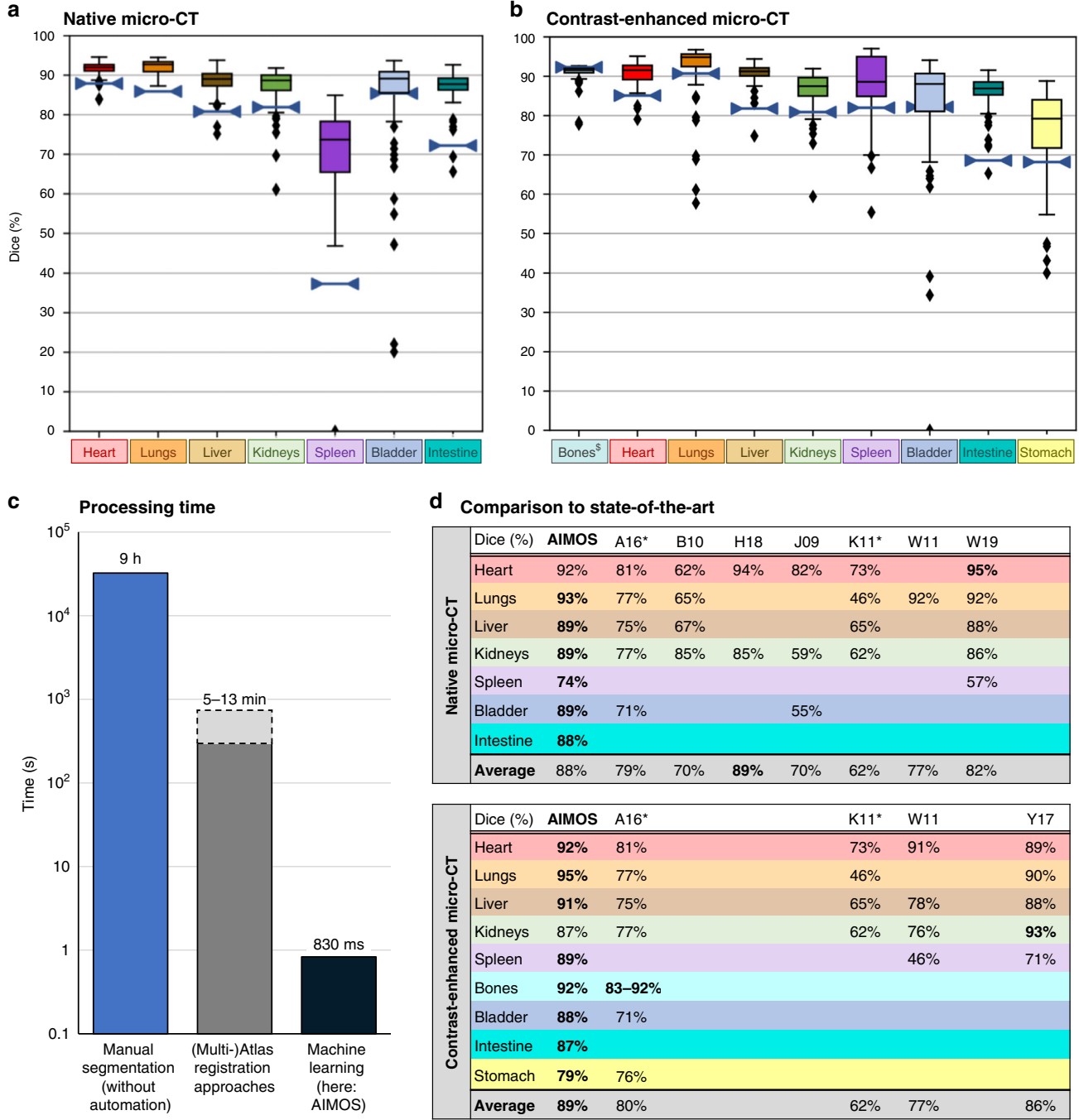

**Fig. 2 Quantitative results of segmentation performance. a**, **b** Box plots of Dice scores per organ; each box extends from lower to upper quartile values of data, with a black line at the median; the whiskers extend to the outmost data point within 1.5× the interquartile range; the outliers beyond the whiskers are shown as diamonds; the blue lines represent human segmentation performance. ${}^{\$}$The human annotators segmented bones with a semi-automatic thresholding function, which may overestimate human consistency as compared to purely manual segmentation. **a** Each box plot represents $n = 140$ independent scans from 20 biologically independent animals. **b** Each box plot represents $n = 81$ independent scans from eight biologically independent animals. **c** Processing time (in log scale) to segment one 3D whole-body scan of a mouse if done manually, with current state-of-the-art methods (atlas registration[36,37]) or with machine learning approaches (here: AIMOS). **d** Median dice scores of AIMOS compared with those of the state-of-the-art from literature for native and contrast-enhanced microCT (scores are not based on the same dataset). Studies marked with asterisks exploited multimodal inputs of native microCT and MRI; as MRI provides high contrasts for soft abdominal organs, these studies are listed for both tables. Source data are provided as a Source Data file.

Dice scores for most other organs, which are between 87% and 95% for the kidneys and the lungs, respectively. For the contrast-enhanced microCT dataset, we also segmented the stomach, another low-contrast organ in CT, for which AIMOS reached a score of 79%, which is lower than for the other organs but higher

than for the human segmentation (69%). The low segmentation performance of human annotators for the spleen suggests that it is intrinsically hard to segment due to its low contrast in native microCT and its variability in size, shape, and exact location. While AIMOS exceeds prior art and human performance,

it nevertheless dramatically benefits from contrast enhancing agents as well. This becomes especially apparent when assessing segmentation performance over time after contrast agent injection. As shown in Supplementary Fig. 3, the spleen can be segmented at a very high Dice score of 92% once the contrast agent has reached its full effectiveness, significantly higher than immediately after injection ($p < 0.01$, two-sided $t$-test). Just like in the native microCT dataset, the AIMOS prediction performance matches or exceeds that of a human expert (blue lines) for all tested organs. It is noteworthy that both human annotations were not completely independent for bones due to the aid of the same thresholding functions, which may overestimate human consistency.

A more detailed overview of all further performance scores for all organs and all datasets is provided in Supplementary Fig. 4. In short, the Hausdorff distance and the center of mass displacement are mostly around or below 1 mm, providing further indications of high segmentation performance. Furthermore, it shows that the optional ensemble-voting can consistently improve results (1–2 p.p. increase in Dice score) but is not essential to derive good segmentations.

Neural networks are generally known to be fast approaches to image segmentation[51]. Even though the time to train a deep convolutional neuronal network depends on technical settings and the size of the dataset and may be in the order of hours, the inference time of AIMOS to predict the segmentation of one whole-body scan is substantially shorter and very consistent. The complete volumetric segmentation of one whole-body scan took AIMOS 0.83 s (Fig. 2c). This is more than four orders of magnitude faster than manual segmentation (9 h in our case; see Methods section for details) and more than two orders of magnitude faster than commonly used atlas registration-based techniques (5–13 min[36,37]). While inference speed benefits from the use of a GPU, the AIMOS has low computational requirements and can also be run on a standard CPU. Thus, while semimanual interactive segmentation tools can reduce the time for expert annotation, machine learning approaches like AIMOS can accelerate organ segmentation tasks substantially without a compromise on quality.

While the experimental setup, the organs of interest, and the acquisition varied, a comparison to reported performance metrics of prior studies indicates that AIMOS matches or exceeds the performance of these segmentation methods despite the dramatically reduced processing time. In Fig. 2d, we compared AIMOS, separately for native and contrast-enhanced microCT, to eight other automated segmentation approaches (abbreviated A16[38], B10[34], H18[37], J09[33], K11[35], W11[36], W19[40], and Y17[39]). Please note that these studies report their performance scores based on different datasets, which are mostly not publicly available. However, they all represent whole-body CT scans of mice. While all of them entail native or contrast-enhanced microCT, A16[38], and K11[35] are multimodal approaches that make use of native microCT and MRI, which has a higher contrast for soft abdominal organs, and are thus also compared with contrast-enhanced CT. Also, please note that A16[38] does not segment the entire skeleton but rather selected bones only. For native microCT, AIMOS obtained good results for the heart (92%), close to the best result (95% for W19[40]). For the lungs, liver, kidneys, spleen, bladder, and intestine, AIMOS reached the highest Dice score reported so far. The overall average of 88% is the second highest, only exceeded by H18[37], that only segmented two organs and stated explicitly that they had decided to exclude organs in low-contrast body regions (such as the spleen and liver) from their study. For contrast-enhanced microCT, AIMOS reached good results for the kidneys, although they were slightly below Y17[39]. For all other organs (the heart, lungs, liver, spleen,

bladder, intestine, and stomach) and the bones, AIMOS yielded the highest Dice score reported so far. Also, the overall average score of 89% is the highest reported so far. In summary, these results show that AIMOS matches or exceeds state-of-the-art and human performance in both datasets.

Figure 3 shows representative samples of the predicted organ segmentations and compares them to the human reference annotation. Visual assessment indicates that the predictions not only closely overlap with the reference, but also seem anatomically plausible. For instance, no implausible predictions such as disconnected structures occur. However, there is of course some deviance from the reference annotation. This prediction error can be broken down into three categories: a) cases with actual errors that show shortcomings of the model, b) cases in which the model yields good predictions but the human reference seems erroneous, and c) cases in which the true segmentation may be hard to judge but where the prediction seems plausible despite a difference from the reference. Examples for b) can be seen for the lungs in Fig. 3a and for the liver in Fig. 3b. Here, AIMOS seems to provide a reasonable prediction but the human reference annotation may be slightly imprecise. This causes lower performance scores despite good predictions. A more in-depth analysis showed that some of the lowest reported performance scores (for all performance metrics) indeed reflected incorrect reference segmentations in the public CT dataset (see Supplementary Fig. 5). Interestingly, in one of these cases the predicted segmentation for the bladder did not overlap at all with the human reference; however, a review of this scan revealed that the human expert misplaced the label for the bladder in this case, an error detected by AIMOS (see Supplementary Fig. 5c).

The most important observation, however, is cases from category c), in which the prediction seems plausible despite being different from the reference and in which it may be hard to judge from the data, where the true delineation should be. Examples can be seen for the kidneys and spleen in Fig. 3a. We speculated that the acquired imaging data represent insufficient information and may thus lead to ambiguous and subjective human interpretation; thus, we decided to explore this with an additional set of experiments.

**Ambiguity of human interpretation of biomedical data.** In order to assess the subjectivity and bias in human interpretation, we analyzed how two human annotations differ and how the AIMOS prediction compares to one reference annotation versus the other. Please note that AIMOS was trained on segmentations from annotator #1 (the predictions were always made on scans spared during training; see Methods section for details). With this experiment, we aimed to test whether AIMOS would be able to learn the individual interpretation of annotator #1. Indeed, we saw subjective, diverging interpretations of the same data (Fig. 4a, b) when comparing independent reference segmentations by two human experts. The prediction by AIMOS seemed to match annotation #1 rather closely but differed from annotation #2, here shown for the spleen (Fig. 4a) and for the kidneys (Fig. 4b).

To test the hypothesis that AIMOS learns to follow the subjective interpretation of annotator #1, on whose data it was trained, we systematically analyzed this for both datasets (Fig. 4c, d). We observed a consistently higher Dice score between AIMOS and annotator #1 than for both, the Dice scores between both annotators and between AIMOS and annotator #2. Note that the human segmentation of the bones was done with a semiautomatic thresholding function, which may overestimate human consistency as compared to purely manual segmentation. These findings suggest that AIMOS is indeed capable of learning the individual interpretation of a given annotator (here: annotator #1) and that a

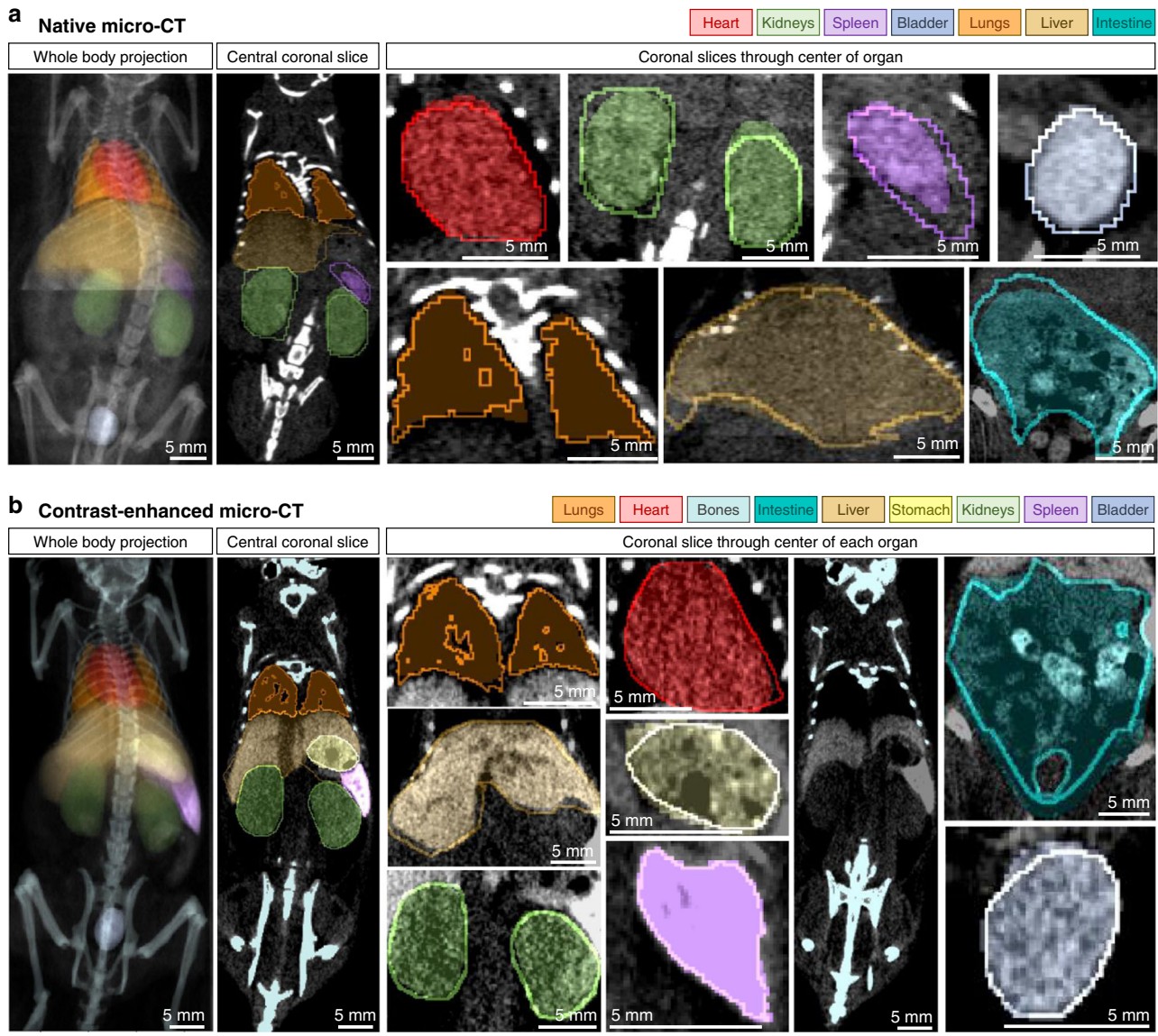

**Fig. 3 Qualitative results of segmentation performance.** Filled, semi-transparent areas show AIMOS prediction for **a** native micro-CT and **b** contrast-enhanced micro-CT. Opaque lines indicate human expert annotations. Left column shows mean-intensity projections for the whole-body scan. Second column shows representative coronal slices. Remaining columns of figure show individual organs (coronal slice through center of organ). The intestine segmentations are not displayed in the whole-body projections as they would occlude other organs due to their size and location.

second, independent annotation is needed to assess the generalization performance of any such model. Depending on which annotation is defined as the reference, AIMOS exceeds human performance or AIMOS matches human performance.

**AIMOS predicts the regions of diverging human interpretation.** Motivated by this observation, we speculated that AIMOS could then also learn to predict those regions in the image, where human interpretation may diverge due to ambiguity in the data. To test this hypothesis, we modified the setup of AIMOS in order to reflect the agreement and disagreement of human annotations separately (Fig. 5a). Also in this setup, AIMOS predicted organ segmentations with high overlap to the agreed-upon reference annotation (Fig. 5b). But importantly, it can now also predict a heat map of image regions in which the human annotators are most likely to disagree (Fig. 5c), which closely reflects their actual disagreement (Fig. 5d). A quantitative analysis of the prediction of disagreement was performed by binarizing the heatmap (threshold at 50%), which allowed to determine the Dice score

with the actual disagreement. This score was 55% for the contrast-enhanced and 60% for native microCT dataset. Given that per definitionem there are no clear boundaries of regions, where human annotators disagree these scores are high and thus indicate that the AIMOS prediction indeed closely reflects the actual disagreement. Interestingly, the predicted heat map did not just merely follow the outlines of the organs, which could be explained by small differences in segmentation caused by slightly imprecise delineations. The heat map seemed to be unevenly distributed, suggesting a high degree of agreement in some regions (e.g., outline of the lungs), and a very low degree of agreement in low-contrast regions such as the caudal parts of the liver and the entire region around the spleen.

If we re-evaluate AIMOS on the basis that a prediction is deemed correct if it matches at least one of both annotations (voxel-by-voxel based decision), the Dice scores increase substantially (Fig. 5e). For native microCT, the score for the spleen increased by 11 p.p. to 86% and all other organ segmentations also reached very high scores between 94 and

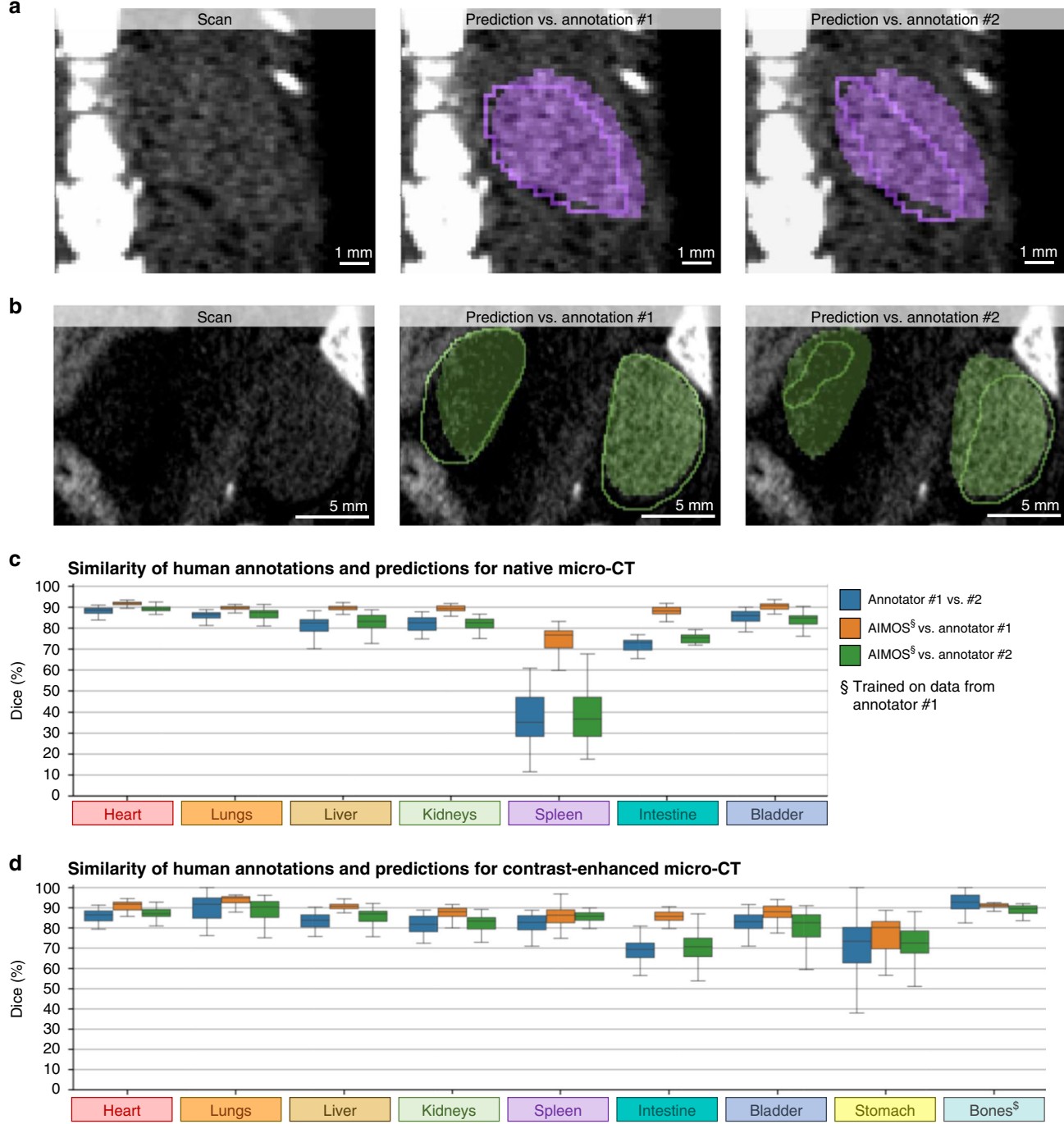

**Fig. 4 Human annotation may reflect an individual expert's bias. a** Raw scan (left) of a spleen in native microCT, the predicted segmentation overlaid with annotation of human expert #1 (middle), and overlaid with annotation of human expert #2 (right). **b** Same as **a** but for kidneys in contrast-enhanced micro-CT. **c**, **d** Dice score for segmentations by expert #1 vs. expert #2 (blue), AIMOS vs. expert #1 (orange), and AIMOS vs. expert #2 (green). Each box extends from lower to upper quartile values of data, with a black line at the median; the whiskers extend to the outmost data point within 1.5× the interquartile range. **c** For the native CT dataset, each box plot represents $n = 35$ independent scans from five biologically independent animals (all that were annotated twice). **d** For the contrast-enhanced CT dataset, each box plot represents $n = 38$ independent scans from eight biologically independent animals (all that were annotated twice). Note that AIMOS was trained on annotations from expert #1. $^\$$The human segmentation of bones was done with a semiautomatic thresholding function, which may overestimate human consistency as compared to purely manual segmentation. Source data are provided as a Source Data file.

96%. The same applied for contrast-enhanced microCT, for which all scores were between 94 and 97%. These findings indicate that while the predicted segmentations do not exactly follow one single, given reference annotation, they do not deviate more from this reference annotation as an independent human expert would.

**Quantitative application of AIMOS for uncertainty quantification.** However, the fact that human interpretation of the same data may diverge to the extent described above has far-reaching consequences in a broad range of biological questions. Here, the capability of AIMOS to predict areas of human disagreement finds direct application in quantifying the resulting uncertainty of

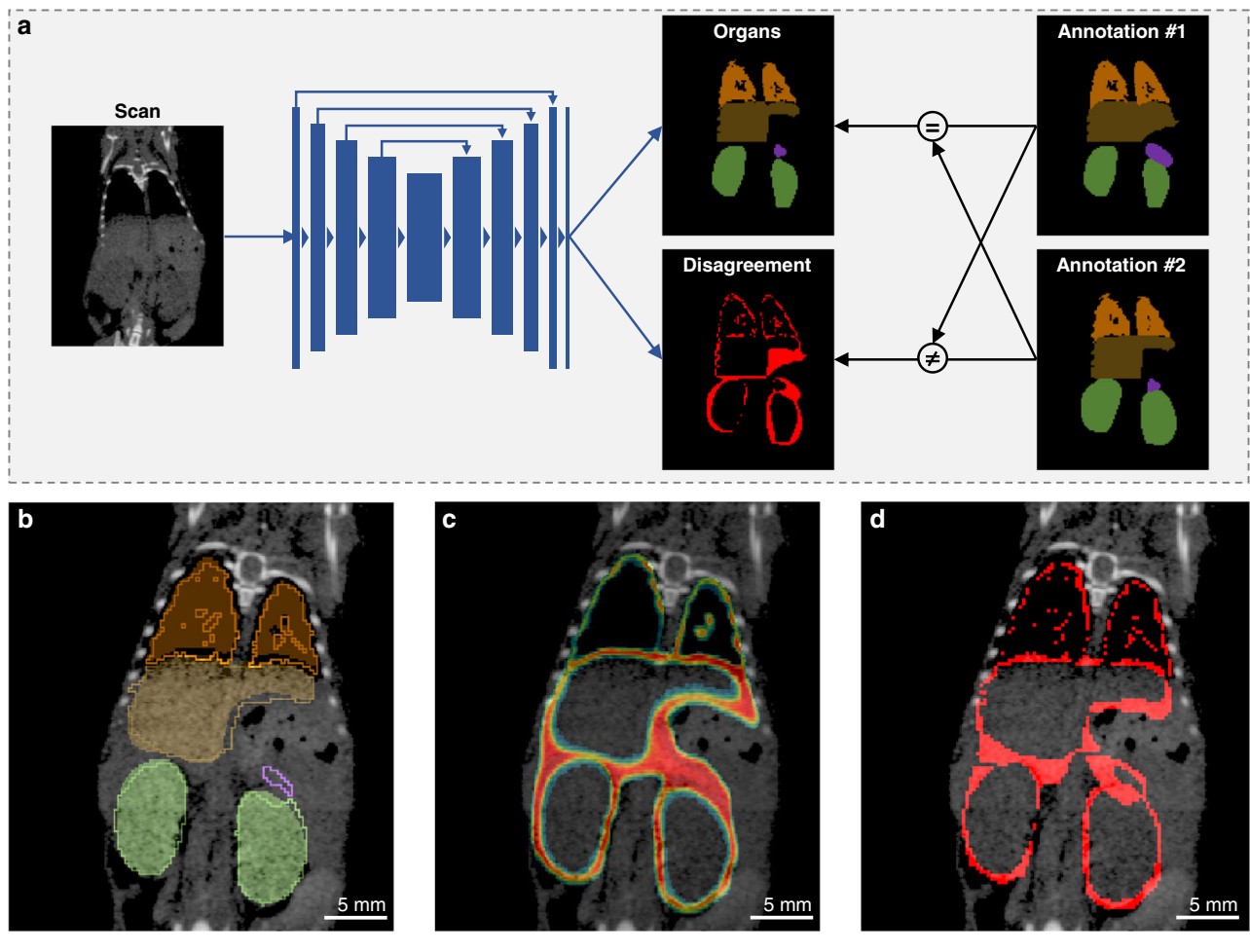

**Fig. 5 AIMOS learns to predict areas of likely human disagreement. a** Workflow for training procedure. Making use of both available human expert annotations, AIMOS is trained to predict not only the location of all organs but also areas of human disagreement. **b, c** Example of an AIMOS prediction on test data **b** for organ segmentations (overlaid by a thin line for human labels) and **c** for likely human disagreement. **d** Actual human disagreement (difference of both human expert annotations). **e** Performance of AIMOS prediction of organ segmentations (Dice score) as compared to that of the reference from annotator #1 (left column) and as compared to that of references from both annotators (right column). **f** Organ volumetry for both datasets. Left column shows average volume per organ across mice; other columns show variability (standard deviation relative to average volume) across mice (intersubject), across annotations from two annotators for the same scan (interannotator), and across annotations for the same mouse from the same annotator (intraannotator). $Semiautomatic thresholding aided human segmentations of bones and may thus underestimate intraannotator and interannotator variability.

organ segmentations. For example, for the field of organ volumetry, in which the volume and the shape of organs between test and control groups are compared in order to draw conclusions about physiological and pathological developments. Assessing whether a difference between both groups is statistically significant of course requires taking the variance within the groups into account (intersubject variability). But if manual segmentations do not represent ground truth and can reflect subjective

interpretation and bias, this must also be taken into account. In order to explore the relevance of this, we quantified the magnitude of three sources of variability for both datasets (Fig. 5f): a) intersubject variability, the natural anatomical variability from mouse to mouse within each dataset; b) interannotator variability, the variability of annotations from independent annotators for the same scan; and c) intraannotator variability, the variability of several annotations for the same mouse from a single annotator. We observed an average intersubject variability of 16%. However, the interannotator variability was substantially higher (26–37%). Importantly, this was also substantially higher than the intraannotator variability (14–15%), suggesting that the interannotator variability was mainly driven by individual bias, and subjective interpretations due to ambiguous data. Altogether, these findings highlight the relevance of appreciating the intrinsic ambiguity in organ segmentations. AIMOS achieves this by automatically quantifying and localizing uncertainty in the data. This enables more consistent segmentations and explicit decisions how to treat ambiguous regions for further analysis.

**Applicability to other imaging modalities**. In a last experiment, we aimed at assessing the versatility of AIMOS for other imaging modalities and for direct biomedical application. Here, we chose volumetric light-sheet microscopy of cleared mice, a modality that substantially differs from microCT in many aspects (Fig. 6a). As described in the Methods section, mouse bodies were first rendered transparent using tissue clearing and then scanned with a light-sheet microscope in 3D at a high, isotropic resolution of 6 μm. Signal acquisition was performed at two different wavelengths, one capturing the native image signal (auto-fluorescence) and the other capturing the fluorescent signal of PI, which stains cell nuclei. We acquired a total of 15 whole-body scans from 15 different mice. Scans were annotated manually by human experts for the brain, heart, lungs, liver, kidneys, and spleen.

Figure 6b shows the quantitative assessment of the prediction performance of AIMOS. Despite the substantially smaller training data size, the Dice scores were very comparable to the performance on the microCT datasets. Again, the spleen was segmented at a comparatively low Dice score (with higher variance) of 73%. The Dice scores for the brain, heart, liver, lungs, and kidneys, however, ranged between 83% for the liver and 89% for heart and kidneys with very small interquartile ranges. Importantly, AIMOS showed a very similar performance on the light-sheet microscopy dataset regardless of the variant (average performance for native light-sheet microscopy: 84%; average performance for nucleus-staining fluorescence microscopy: 86%). In addition, a qualitative assessment suggested that the predicted segmentations are anatomically plausible (Fig. 6c). Again, the organs were localized correctly and while some differences around the fine details of the outline can be seen (e.g., for the brain and liver), the overall organ shapes were well predicted. A comparison to the other, substantially larger datasets indicated that the performance may improve further with additional training data (Supplementary Fig. 1c). Nevertheless, this analysis suggested that AIMOS can already be efficiently trained from scratch with as little as 8–12 annotated scans. Importantly, the need for annotated training data was further substantially reduced by using pretrained models (which we provide), even when modalities differed (Supplementary Fig. 1d–f).

To assess the precision of organ delineations in the practical settings of biomedical applications, we let AIMOS segment a scan of a tumor-bearing mouse from a prior study[17] in order to quantify the spatial organ distribution of metastases (Fig. 6d). Metastases were counted for each organ based on the automatic segmentation provided by AIMOS and based on a manual segmentation by a human expert (see Methods section for details). A comparison with the true metastasis counts revealed that AIMOS-derived counts consistently matched or exceeded human precision. This suggested that, despite the small number of training samples, the quality of predicted organ delineations were sufficient for direct biomedical application of AIMOS. Altogether, these findings suggest that the AIMOS pipeline, without any modification or tuning, can be readily applied to fundamentally different imaging modalities and trained successfully with very small training datasets.

## Discussion

In this study, we introduced AIMOS, a simple and lean deep learning-enabled pipeline for organ segmentation in whole-body scans of mice. Delineating organs manually is not only a tedious task, but also inherently difficult, especially in low-contrast modalities such as microCT, even for experts. Thus, there is a great need for reliable, automated methods to derive high-quality organ segmentations. In contrast to commonly used atlas-registration-based[33–37] and a recent learning-based approach[40], AIMOS represents an end-to-end deep neural network that directly segments a large number of organs in mouse scans. This approach works without human intervention, processes a full scan in less than 1 s, and delineates the organs and skeleton with high precision. We evaluated the model performance in the largest validation experiment conducted so far, with over 220 scans from a public dataset[28]. The reported segmentation quality matches or exceeds state-of-the-art and, importantly, the quality of human expert annotations. The approach is not tied to a specific imaging modality and is not sensitive to parameters such as imaging resolution, highlighting its broad applicability. The entire dataset including annotations, the code, and the trained models are publicly available to enable direct application and foster further improvement and adaptation. As shown in the light-sheet microscopy dataset, the model can be easily retrained from scratch to be used in other modalities with as little as 8–12 scans and our pretrained models reduce this need even further.

Motivated by the observation that AIMOS can exceed a human expert in predicting a given reference annotation, we assessed the sources of variability in human annotations. We showed that AIMOS can learn to follow the subjective interpretation of a given expert. In addition, we showed that the interannotator variability is substantially larger than the intraannotator variability. Both findings suggest that expert annotations are subject to human bias in interpretation, especially when there is ambiguity in the acquired imaging data, an issue previously reported[45,52]. This has intricate effects on the informative value of performance metrics such as the Dice score if only a single annotation is used as a reference, which is a common practice in the field[34–40]. We showed that AIMOS exceeds human performance when one expert is defined as the reference but matches human performance when another expert is defined as the reference. Thus, while measuring the Dice score against a single human annotation may be helpful for model development, only a second, independent annotation as a test reference reveals its true generalization performance relative to human experts.

Future work may explore to further increase robustness and generalizability. Potential avenues include the use of multiple, more diverse datasets (for example, acquired at different labs and acquired with different modalities), 3D convolutions (which may help with complex structures such as the spleen but also require substantially more training data for convergence), and combinations with reference atlases (for example, to detect and flag unlikely predictions). However, the observation that human

**Fig. 6 AIMOS can be applied to other imaging modalities. a** 3D light-sheet microscopy dataset was obtained from 15 mice at a resolution of 6 μm/voxel. Each scan is available without staining (native) and with fluorescent nucleus-staining with propidium-iodide (PI). **b** Box plots of Dice scores per organ; each box extends from lower to upper quartile values of data, with a black line at the median; whiskers extend to outmost data point within 1.5× the interquartile range; outliers beyond whiskers are shown as diamonds; each box plot represents represents $n = 15$ independent scans from 15 biologically independent animals. **c** Filled, semitransparent areas show AIMOS prediction, opaque lines indicate human expert annotation. Left column shows average intensity for whole-body scan; second column shows representative coronal slice. Right half of panel shows individual organs (coronal slice through center of organs). **d** Bar charts and mean-intensity projections of metastasis counts per organ, based on segmentation by AIMOS or a human expert (a detailed description of this visualization is provided in the Methods section). Location of metastases indicated with circle (diameter not to scale); metastases are shown in gray if they were incorrectly allocated to the organ based on imprecise segmentation by AIMOS or human. Green line in bar charts indicates true metastasis count. Percentage above bar charts represents relative error in metastasis count. Source data are provided as a Source Data file.

experts may disagree substantially in their data interpretations draws the usefulness of further model development just for higher Dice scores into question. Rather than matching one reference to an ever higher degree, investigating and localizing sources of ambiguity may be an important aspect of future research. Here, we trained AIMOS to segment organs where both annotators agree and to identify image regions, where annotators are most

likely to disagree. This enables researchers to either discuss those ambiguous regions in detail in order to find a consensus, or to appreciate this intrinsic uncertainty for any further statistical analysis. Here, we showed that the interannotator variability can be of substantial magnitude, exceeding the intersubject variability (often the only source of variability taken into account[15,53]). This highlights the relevance of factoring in this uncertainty for further

analysis, for example, when assessing the significance of differences between control and test groups of mice.

AIMOS as an enabling resource will be helpful in many areas of biomedical research, including tumor research[15–17,54], organ lesion studies[18], drug delivery[20], and nanoparticle uptake[22–26]. It provides high-quality organ segmentations within a second. It has low barriers for adoption, has low computational requirements, and requires no human intervention or parameter tuning. In addition, it does not act as a black box but provides probability maps for each organ and sheds light on ambiguous image regions requiring further discussion. AIMOS helps to automatically quantify this uncertainty and thereby aids in refining statistical analyses. In summary, AIMOS as a tool and the findings of this study are an important contribution to increase scalability, reduce bias, and foster reproducibility in many areas of biomedical research.

## Methods

**Data**. We trained and evaluated AIMOS on four datasets from different imaging modalities and variants: native microCT, contrast-enhanced micro-CT, native light-sheet microscopy, and nucleus-staining fluorescence light-sheet microscopy. The CT dataset is publicly available and was explicitly designed to serve as a basis for deep learning-based models for mouse organ segmentation[28]. All details on the data acquisition and the annotation process can be found in the data descriptor[28], and will thus not be repeated here beyond key information and further processing steps. For both datasets, the mice were scanned several times. Before each scan, they were newly anesthetized and newly positioned in the CT bed, resulting in different postures of the same mouse in different scans. For the native micro-CT dataset, all available 140 scans from 20 mice were included in this study. While most scans have a common field of view from the shoulder to the hips, a small number of scans (#16–#20) had a larger field of view that included the head. These scans were cropped to match the common field of view. No further preprocessing was applied to this dataset. For the contrast-enhanced microCT dataset, the mice were injected with an agent to increase the contrast of the liver and spleen. All scans have a common field of view and no cropping was necessary. However, out of 85 scans from 10 mice, only 81 scans from 8 mice were included in this study. Two scans were excluded due to missing annotations (M01_48h and M05_144h). Two additional scans (M09_024h and M10_024h) were excluded as they were the only scans without the time series that was provided for all other mice and is needed for this study. Some scans contained motion artefacts or other acquisition artefacts[28], but were nevertheless fully included in order to reflect the full range of the data and any shortcomings. While both datasets were largely annotated manually, semi-automated thresholding was applied to aid the segmentation of the bones. Importantly, a subset of both datasets was annotated a second time by an independent expert (following the same protocol), allowing the assessment of inter-annotator variability in human expert segmentations. The provided segmentations were not altered (despite some minor segmentation inaccuracies, see "Results" section) to maintain reproducibility. To the same end, the nomenclature of the time points was not altered; however, please note that scans with negative time points ($t = -1$ h) were also obtained after injection of the CT contrast-agent.

The light-sheet microscopy dataset was partially acquired for a prior study[17] but was extended for this study. This imaging modality enables whole-body scans at microscopic resolution (6 μm) for entire mice without the need for destructive cryoslicing. Volumetric light-sheet microscopy requires clearing of (and optionally the application of nucleus-staining fluorescent PI in) mouse bodies. The entire procedure, including image acquisition, has been described in prior work[17,55–57] and will thus not be repeated here beyond key information and further processing steps. We used mice from the following strains: NSG (NOD/SCID/IL2 receptor gamma chain knockout), NMRI nu/nu mice, C57BL/6 (sex: male and female; age: 1–4 months). All animal experiments were conducted according to institutional guidelines of the Ludwig Maximilian University of Munich, the Technical University of Munich, the Helmholtz Center Munich, the University of Giessen, and the University of Frankfurt. Experiments were conducted after approval of the Institutional Animal Care and Use Committees (IACUC) of Technical University of Munich, Ethical Review Board of Regierung von Oberbayern, the UK Home Office, and the veterinary department of the regional council in Darmstadt, Hesse, Germany. A total of 15 scans from 15 mice were acquired. All scans were cropped to a common field of view, from the head to the hips, which partially excluded some outer parts of the extremities. The brain, heart, lungs, liver, kidneys, and spleen were segmented manually with the ROI Manager of Fiji[58] by closely reconstructing the delineations with polygons in regularly spaced slices, and subsequent interpolation of the delineation between those slices. In a second step, the derived volumetric segmentations were further refined in ITK-Snap[59] to ensure a good fit across the entire volume. The expert annotators took detailed minutes on the progress of the annotations, which served as the basis to calculate the total amount of time to fully segment all organs per mouse scan.

**Model**. The AIMOS pipeline consists of a preprocessing module, a deep learning backbone, and a postprocessing module. The preprocessing module slices the volume into coronal 2D images, normalizes the signal intensity by subtracting its mean and dividing it by its standard deviation, and resamples them to a desired resolution. The sensitivity of the model to the resolution resampling was analyzed in an ablation study (see Results section). During training, only slices that showed at least 1 voxel from 1 organ (as determined by the training annotations) were fed to the network; input images and annotations were augmented by random rotations (±10°) and random cropping (down to 80% of original image area). During inference, all slices of the entire volume were presented for prediction and no augmentation was performed.

The deep learning backbone of the model followed a U-Net-like[60] architecture and consisted of an encoding and a decoding path, that were interconnected with skip connections. While the number of encoding and decoding levels was varied (see Results section for an study on the sensitivity of the model to this), the encoding and decoding units in each level always followed the same structure. Each encoding unit consisted of two convolutions (kernel size: 3; padding: 1; stride: 1), batch normalization, a rectifying linear activation function, and a max-pooling operation (kernel size: 2; stride: 2). Each encoding unit has twice the number of feature channels as compared with the previous one; the first encoding unit has 32 feature channels. Each decoding unit consisted of three convolutions (same parameters), whereas the first one received the concatenation of an upsampled input from the previous level (bilinear interpolation) with the input from the skip connection from the corresponding encoding unit. Each decoding unit has the same number of feature channels as the corresponding encoding unit from the same level. The very last convolution of the model mapped the 32 feature channels to the number of prediction classes (i.e., the number of organs to be predicted). A final pass through a sigmoid function yielded volumetric probability maps of each organ. The model was trained with a soft-Dice loss function and the Adam optimizer[61]. The training and evaluation procedure was designed to assess the generalization performance of AIMOS on unseen data. To this end, we followed a nested k-fold cross-validation procedure that splits the dataset into three distinct sets (a training set for model weight optimization, a validation set for hyper-parameter optimization, and a test set for evaluation). Importantly, the dataset was split on the level of individual mice, not on the level of scans, since the mice were scanned multiple times. This avoids any information leak between two scans of the same mouse across the training and test sets. We chose k to represent the number of mice per dataset. For each given dataset split, the model was trained for 30 epochs on the training dataset and the learning rate was gradually reduced when the performance on the validation set stagnated for five epochs (initial learning rate: $10^{-3}$).

The postprocessing module transforms the volumetric probability maps from the deep learning backbone into the final model prediction. We trained an ensemble of ten models per training/validation set, each time with a different mouse chosen for validation, and chose the median of the predicted probabilities per voxel as the basis for the final prediction on the test set. This ensemble-voting procedure is optional and not needed per se but further improved model accuracy by eliminating outlier predictions. Subsequently, the probability maps for all prediction classes for anatomical segmentation (i.e., all organs and the bones) were binarized with a softmax function. This was not the case for the uncertainty prediction class, as its final output is a heatmap, not a segmentation. Please note that the uncertainty prediction is directly learned from human disagreement in the expert annotations and independent from the ensemble-voting procedure. Reconstruction of predictions in 3D yielded the final output of the AIMOS pipeline. No further postprocessing steps such as filtering or morphological operations were applied.

The entire pipeline was implemented in Python and only required a small number of open-source packages: PyTorch[62] (deep learning framework), SciPy[63–65] (scientific computing), and NiBabel[66] (input/output for volumetric data in the NIfTI file format). On a standard workstation equipped with a GPU (here, we used a NVIDIA Titan Xp), model training for one epoch took between 1 and 3 min (depending on dataset). While inference speed benefits from the use of a GPU, the trained AIMOS pipeline does not require GPU support for mouse organ segmentation and can be run on a standard CPU.

**Analysis**. For quantitative performance evaluation, we measured the volumetric overlap between the predicted segmentation in unseen test data, and the human reference annotation with the commonly used SrensenDice score as well as the Hausdorff distance and the center of mass displacement. This evaluation was performed on the reconstructed 3D volume of the entire scan; we computed one score for each organ in each scan on the basis of the human reference annotation for that organ and that scan.

For the Dice score, a given voxel of the predicted segmentation is classified as either a true positive (TP) if it was correctly predicted to be part of a given organ, as a false positive (FP) if it was incorrectly predicted to be part of a given organ, or as a false negative (FN) if it was incorrectly predicted to be background despite being part of a given organ. The Dice score is then computed as follows (with $\epsilon = 10^{-5}$ for numerical stability):

$$\text{Dice} = \frac{2\text{TP} + \epsilon}{2\text{TP} + \text{FP} + \text{FN} + \epsilon}. \qquad (1)$$

To assess the human performance for comparison, we quantified the similarity of human expert annotations with the Dice score, following the same logic and treating one human segmentation as the reference and the other as a prediction.

Given the symmetry of the approach, both annotations can be interchanged without a change in Dice score.

The Hausdorff distance is expressed as a percentile of the distances between the surface of the predicted segmentation, and the surface of the human reference annotation for a given organ in a given scan. We report the 50th percentile (the median distance) and the 95th percentile. The Hausdorff distance is computed as follows (with $p$ as the percentile, $\hat{S}$ as the predicted segmentation, and $S$ as the human reference annotation):

$$\text{HD}_p = \text{percentile}\left(p; \max_{s \in S}\left(\min_{\hat{s} \in \hat{S}} ||\hat{s} - s||\right)\right). \quad (2)$$

The center of mass displacement represents the distance between the centers of mass (CoM) between the predicted segmentation and the human reference annotation for a given organ in a given scan:

$$\Delta\text{CoM} = ||\text{CoM}(\hat{S}) - \text{CoM}(S)||. \quad (3)$$

For qualitative performance evaluation, the predicted organ segmentations, the human reference annotations, and the scans were overlaid in 3D for visual inspection. To visualize this evaluation for this manuscript, we chose a combination of volumetric projections and cross-sections. All such visualizations presented in this study were generated with the same, predefined approach to avoid bias and ensure objectivity. To determine the position of the cross-section, we always chose the coronal slice through the center of the organ. For whole-body cross-sections, the slice that showed the maximum area with the maximum number of organs was chosen. To provide further information on all other slices of the volume, we additionally displayed mean-intensity projections for the entire volume.

For the assessment of variability in organ volumetry, we quantified the mean volume per organ across mice for each dataset and quantified three kinds of variability. In the following equations, $m$ will denote the index of a given mouse and $M$ the total number of mice of the dataset; $t$ will denote the index of a time point of a scan of a mouse and $T$ the total number of scans per mouse; $a$ will denote the index of a given annotator and $A$ the total number of annotators; and $o$ will denote the index of a given organ.

The intersubject variability quantifies the standard deviation of organ volumes across mice:

$$\text{Intersubject variability}(o) = \sqrt{\sum_{m=1}^{M}\left(\overline{v_{m,o}} - \frac{1}{M}\sum_{m=1}^{M}\overline{v_{m,o}}\right)^2}$$
$$\text{with} \quad \overline{v_{m,o}} = \frac{1}{T}\sum_{t=1}^{T} v_{m,t,o} \quad (4)$$

The interannotator variability quantifies the mean standard deviation of independently created annotations across all scans:

$$\text{Interannotator variability}(o) = \frac{1}{M}\sum_{m=1}^{M}\frac{1}{T}\sum_{t=1}^{T}$$
$$\sqrt{\sum_{a=1}^{A}\left(v_{m,t,o,a} - \frac{1}{A}\sum_{a=1}^{A} v_{m,t,o,a}\right)^2} \quad (5)$$

The intraannotator variability quantifies the mean standard deviation of several annotations by the same annotator for the same mouse:

$$\text{Intraannotator variability}(o) = \frac{1}{M}\sum_{m=1}^{M}$$
$$\sqrt{\sum_{t=1}^{T}\left(v_{m,t,o,a=1} - \frac{1}{T}\sum_{t=1}^{T} v_{m,t,o,a=1}\right)^2}. \quad (6)$$

To analyse the precision of the organ delineations derived by AIMOS from the perspective of a biomedical application, we quantified the spatial distribution of cancer metastases in volumetric light-sheet microscopy scans. To this end, we used a mouse scan acquired for a prior study[17], in which metastases were identified automatically with a deep learning algorithm and subsequently allocated to organs manually. Here, we recreated the organ allocation for each metastasis based on the volumetric organ segmentations by AIMOS and by a human expert annotator and compared the allocations to a manually verified ground truth allocation. For both allocations, we quantified the metastasis count per organ and the relative error to the true count. To visualize the allocation of metastases for each organ, the volumetric scan of an organ and the AIMOS segmentation was projected along the dorsoventral axis (mean-intensity projection). The true delineation was indicated with a solid line along the outline of the projected human reference annotation (please refer to the Data section for a description of the annotation process). Please note that this kind of visualization displays the outermost extent of the contour of the human reference annotation from that perspective; given that the imaging data itself is displayed with a mean-intensity projection, the outermost part of that organ may sometimes appear rather dim. The location of metastases were indicated with circles (diameter not to scale); color-coding indicated whether a metastasis truly belonged to an organ or was incorrectly allocated to the organ (gray) due to an imprecise segmentation provided by the human expert or AIMOS.

**Reporting summary**. Further information on research design is available in the Nature Research Reporting Summary linked to this article.

## Data availability
All the imaging data and corresponding annotations used in this study are open source and freely available online. Both the micro-CT datasets (native and contrast-enhanced) including annotations are available at Nature Scientific Data[28]. We deposited the light-sheet microscopy dataset (native and with nucleus-staining fluorescent signal from PI) including annotations and all pretrained models as public datasets on the Harvard Dataverse (https://doi.org/10.7910/DVN/LL3C1R[67]; https://doi.org/10.7910/DVN/G6VLZN[68]). All further relevant data are available from the authors. Source data are provided with this paper.

## Code availability
All the code and trained models to apply, recreate, and modify AIMOS are open source and freely available online. The AIMOS network architecture (including its variants) and the code for model training and inference are made available on GitHub: https://doi.org/10.5281/zenodo.4048770[69]. Please also find a fully functional online demonstration on CodeOcean: https://doi.org/10.24433/CO.2308253.v1[70]. Source data are provided with this paper.

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

## Acknowledgements

We thank Shan Zhao, Ilgin Kolabas, and Diana Waldmannstetter for helpful discussions throughout the project. This work was supported by the German Federal Ministry of Education and Research via the Software Campus initiative (to O.S.). A.E. was supported by Vascular Dementia Research Foundation, Deutsche Forschungsgemeinschaft (DFG, German Research Foundation) under Germanys Excellence Strategy within the framework of the Munich Cluster for Systems Neurology (EXC 2145 SyNergy ID 390857198), Fritz Thyssen Stiftung (Ref. 10.17.1.019MN), and DFG research grant (Ref. ER 810/2-1). NVIDIA supported this work with a Titan XP via the GPU Grant Program.

## Author contributions

O.S. and B.H.M conceived and initiated the project. O.S. led the model development, performed data analysis, and wrote the manuscript. C.P. acquired the light-sheet microscopy data and helped with data interpretation. C.P., H.M., Z.R., and A.M. annotated the light-sheet microscopy data. M.T. helped with data processing. J.C. and F.N. helped with model development, H.L. helped with model evaluation. A.E. and B.H.M. provided guidance throughout the project and edited the manuscript. All authors reviewed the manuscript.

## Funding

## Competing interests

The authors declare no competing interests.
