## [Peer Review File · Nature Communications]

Reviewers' Comments:

Reviewer #1:

Remarks to the Author:

The manuscript by Oliver Schoppe et al. presents a deep learning method for organ segmentation called AIMOS that can perform automatic segmentation of brain, lung, heart, liver, kidney, spleen and the skeleton. The authors demonstrate applications of the AIMOS method, for example an analysis of the distribution of cancer metastases. They show that AIMOS can identify regions in an un-biased manner without typical human errors. AIMOS was trained and evaluated on publicly available micro-CT datasets, native micro-CT and contrast-enhanced micro-CT whole-body scans, and light-sheet microscopy datasets. The authors conclude that AIMOS is a powerful open-source tool to increase scalability, reduce bias, and foster reproducibility in many areas of biomedical research.

The manuscript is well-written, and the figures are clear and easy to understand. Although I am excited about the results presented, I have some concerns that the authors should address before becoming convinced of AIMOS's novelty, strength and utility.

Major comments

1. The authors compare the AIMOS method with many other deep-learning tools and they write that "learnable convolutional kernels provide the most promising basis for automatic organ segmentation". But it is not clear to me what new benefits and breakthroughs are offered by the AIMOS method. In fact, reference 42 shows that the W19 method has very similar performance as AIMOS. What are the main benefits of AIMOS? It seems that the training used for AIMOS is quite much greater than for W19 (TS-DSN) (140 vs. 40). Is it possible that the larger training performed by AIMOS is the reason why AIMOS performs better than TS-DSN? Indicate better benefits and breakthroughs offered by the AIMOS method.
2. I think there are some statements that are exaggerated and misleading that should be reworded. For example, line 235, "While AIMOS exceeds prior art and human performance by a margin ..." and line 18, "with unrivalled accuracy".
3. The introductory section does not mention why this work is important and how the AIMOS method can contribute to science. What are the weaknesses and limitations of existing segmentation methods? How can AIMOS bridge this gap?
4. In this manuscript, the AIMOS segmentation method is challenged with micro-CT datasets. What should be emphasized is that micro-CT images have quite low contrast and are difficult to segment, even for human experts. But this fact is mentioned only for the spleen and that the human annotators had low segmentation performance for the spleen. The authors need to explain this better to sell their story to a wide readership. If information about the difficulties in segmenting micro-CT data sets, even for human experts, is mentioned, it strengthens the AIMOS method.
5. The authors state that the AIMOS method is orders of magnitude faster than prior algorithms. Processing times should be compared to other methods that use deep learning, not just compare it with manual and atlas registration methods. It is already well known that methods based on machine learning are much faster than manual segmentation and the registration method.
6. Figure 6 shows a full-body scan of a mouse recorded with a light sheet microscope. The light-sheet microscopy images have higher contrast than the micro-CT images, and AIMOS can correctly segment organs from light-sheet data. However, the "human expert" appeared to have difficulty selecting bodies in the light-sheet images; the human expert chose the larger area with a black

background. This can lead to poor scores for metastatic quantification of the human expert. Then it will be too much to say that AIMOS exceeds the segmentation quality for human experts. Did the human experts really do their best segmentation? Describe how the segmentation was performed.

Minor

Line 18, The phrase "unrivalled accuracy" in the abstract feels a too strong statement.

Line 29, The start of paragraph two in the Introduction feels abrupt. Maybe the last sentence of paragraph one can go to the beginning of paragraph two.

Line 200 and Figure 1, change "fluorescent light-sheet microscopy" to "fluorescence light-sheet microscopy"

Figures 1 and 6d, Supplementary Figure S3f,g,h, add scale bars.

Figures 4a,b, 5b,c,d, Supplementary Figure S2, please add scale bars in all panels. Scale bars are present in all other panels, even though they have the same size as neighboring panels, e.g., Fig 3a and b.

Line 219, change "...we the compared..." to "...we then compared...".

Figure 2a, 2b, 6b, S3a-d, S4a, the numbers on the y-axis are too small.

Figure S3e-h, add a space before and after "=".

Figure S4b, "t" should be in italic.

Reviewer #2:

Remarks to the Author:

The authors present a software tool for fully automated mouse organ segmentation for whole-body scans, e.g. from μ CT. Such organ segmentation is frequently required in preclinical research and an accurate and robust automated method would save manual segmentation effort and remove a large potential source of bias because manual segmentation suffers from substantial inter-reader variability. The authors use a large freely available μ CT data base, recently published for this purpose, including scans with and without contrast agent. The authors claim superior segmentation performance compared to previous approaches. Furthermore, they show that annotations from more than one person should be used to better evaluate the performance of automated methods which is a relevant finding. The code is published to enable reproduction of the results which is appreciated. The study, including the software tool, is a highly relevant contribution to the applied imaging community but I recommend several improvements of the manuscript.

In the abstract, it would be good to highlight the aspect that manual annotations from multiple humans should be used to evaluate automated segmentation methods.

In the introduction, manual segmentation by delineation in each slices is mentioned. In many applied studies, interactive segmentation is actually used which includes operations like region growing and cutting. This strongly reduces the time for irregularly shaped organs like lungs and bones which have a strong contrast to their surrounding. In that light, the mentioned duration of 9 hours for manual segmentation seems to be too long as assumption. In our studies we take

around 30 minutes per scan using interactive segmentation. Nevertheless, an automated segmentation would be highly appreciated.

Please clarify that the segmentation scores from other approaches are based on other data sets and therefore not entirely comparable because CT scans can have strongly varying quality. It would be highly interesting to see the performance of some of the other approaches on the same (new) data set.

In page 9, line 303, please clarify how the prediction is evaluated against both annotators. Is this done on a voxel-based decision?

Results would be interesting for the urinary bladder which humans are quite good and fast at segmenting. Our own unpublished automated approaches failed miserably unfortunately for the bladder. The published data base also contains manual annotations of the bladder. In case the bladder turns out to be difficult for AIMOS, please discuss why this is the case and how it could be resolved in future work.

Please clarify that one dice score was computed for each organ, using only the segmentation of that organ.

In the formulas for the variability, the time index should either start at 1 like the other indices or the superscript of the summation operator should be $T-1$ instead of T .

For the contrast-enhanced scans, there seems to be a misunderstanding what pre-injection means. In the used data-based, pre-injection time point means that it was before fluorescent contrast agent injection, i.e. after μ CT contrast injection. That explains why the spleen is visible in the so-called pre-injection scans.

I am not sure if bio-distribution is the right term for cancer metastases. I would rather call it organ-distribution. Bio-distribution is typically used for the distribution of an injected probe.

Please discuss future work, e.g.

-Using a 3D network instead of 2D-networks per slice. This could help for complex 3D shapes where a single slice is providing insufficient information. And it could avoid flickering between slices.

-Could combination with an atlas-based method further improve the results?

Reviewer #3:

Remarks to the Author:

I Summary

The manuscript employs a U-Net architecture to perform mouse whole-body segmentation from both micro-CT and light sheet microscopy volumetric images. The system is evaluated on a large set of human-annotated data in terms of DICE overlap. An additional "uncertainty" output channel is used to predict ambiguous regions.

II Overall Comments

Clarity

The manuscript is very well-written and structured and very simple to follow. The figures are clear and concise. The technical content is simple to grasp.

Originality

The U-Net architecture is well-established and heavily used for various segmentation tasks. The AIMOS system is a slight modification of this architecture. The additional uncertainty channel is an interesting and useful add-on. Overall, the contribution of the paper is the application of a known architecture to a dataset relevant for research.

Significance

The proposed system has the potential to become a standard tool for mice research and quantification due to its simplicity, accuracy and speed. However, there are more labels in the dataset(s) [36, 37]. The impact of the system could be significantly increased if further (more fine-grained) labels such as left/right eye, left/right kidney [36] and brain, tumor, stomach, vena cava, bladder [37] as well as further skeleton labels such as upper/lower limbs, ribs, sternum central [37] would be included. Furthermore, a more exhaustive analysis of the uncertainty prediction and different performance measures would help to better understand strength and weaknesses.

Reproducibility

The data is public and the code is shared on github. The capsule on CodeOcean allows to interactively run the model. So the work can be considered fully reproducible.

Empirical analysis

The manuscript provides an analysis of the segmentation accuracy in terms of DICE score with respect to two human annotators and a comparison to 8 different segmentation algorithms on micro-CT data. Furthermore, an assessment of the validity of the ambiguity prediction is provided and results for light-sheet microscopy are discussed.

a) While the analysis is concise and exhaustive, some aspects could be improved. There are accuracy measure complementing the DICE score e.g. [36] reports Hausdorff distance percentiles and center of mass displacement. An additional analysis with different performance metrics would make the paper much stronger.

b) An ablation study for the slice-wise preprocessing with intensity normalization and resampling should be provided to establish that this is a required step.

c) Similarly, the effect of the ensembling and median smoothing over ten models should be properly analyzed. How much does the performance depend on this? How strongly is the runtime affected?

d) The uncertainty prediction should be analyzed in more depth to establish its value and limits. It would be great to quantify with numbers whether the ambiguity information can be used in practice beyond visual plausibility. For example, how much of the discrepancy between human annotators and prediction falls inside the ambiguity region. Are all voxels which the network classifies as being non ambiguous correctly predicted?

Quality

The manuscript presents an interesting contribution but would benefit from some more empirical assessment.

III Detailed Comments

- 1) The authors report that the AIMOS system is faster and more accurate than previous systems but the precise analysis why the prediction is faster is not given. Is it simply the use of a standard U-Net that is so fast or are there additional tweaks required?

- 2) The term "uncertainty quantification" in the title is a bit of a bold claim. The paper does not quantify uncertainty in a proper probabilistic framework, the paper only qualitatively reports that the predicted ambiguity regions are reasonable. The term "quantification" would require a much deeper analysis. A better term would be something along the lines of "ambiguity forecasting" or "sensitivity estimation" instead.

- 3) Is there a special region, why only 6 labels (heart, lungs, liver, kidneys, spleen, bones) were

used? Shouldn't the use of more detailed label information allow to boost the segmentation performance of lumped labels such as "bones" overall?

[36] Van Der Heyden, B. et al. Automatic multiatlas based organ at risk segmentation in mice. *The Br. journal radiology* 92, 20180364 (2018).

[37] Akselrod-Ballin, A. et al. Multimodal correlative preclinical whole body imaging and segmentation. *Sci. reports* 6, 27940 (2016).

REVIEWER COMMENTS – Response letter

In our study we present a deep learning-based processing pipeline termed AIMOS, the fastest and most accurate solution for the automated segmentation of the major organs and the skeleton in volumetric mouse scans. In the revised version, we have added a number of new experiments and supporting analyses to fully address the reviewers' comments. The major improvements of the study include the following:

- 1) Increased number of organs segmented by AIMOS
We extended a large portion of the experiments to an increased number of organs. AIMOS now successfully segments up to 11 anatomical structures at or above human expert performance: brain, heart, lungs, liver, kidneys (now left and right kidney separately), spleen, bones, bladder (new), stomach (new), intestine (new). All findings and claims hold for the newly added organs.
- 2) Additional performance scores
We have analysed the performance of AIMOS not only with the Dice score but now also with the Hausdorff distance and the center of mass displacement. In line with prior results, AIMOS also performs very well along these metrics.
- 3) In-depth empirical assessment
We extended the empirical assessment of the model performance, addressing more explicitly the reviewer's questions about sensitivity to resolution, model architecture, ensemble-voting, and the number of anatomical labels provided.
- 4) Extended analysis of uncertainty quantification
Furthermore, we have extended the analysis of the direct applicability of AIMOS' capability to localise and quantify uncertainty driven by ambiguous imaging data.

Beyond this, we have improved the introduction, the comparison to prior methods, the description of our methods, and the discussion based on the reviewers' suggestions. We would like to express our gratitude to the reviewers for their well-thought-out, detailed, and very valuable comments, which helped us to substantially improve the manuscript.

Reviewer #1 (Remarks to the Author):

The manuscript by Oliver Schoppe et al. presents a deep learning method for organ segmentation called AIMOS that can perform automatic segmentation of brain, lung, heart, liver, kidney, spleen and the skeleton. The authors demonstrate applications of the AIMOS method, for example an analysis of the distribution of cancer metastases. They show that AIMOS can identify regions in an un-biased manner without typical human errors. AIMOS was trained and evaluated on publicly available micro-CT datasets, native micro-CT and contrast-enhanced micro-CT whole-body scans, and light-sheet microscopy datasets. The authors conclude that AIMOS is a powerful open-source tool to increase scalability, reduce bias, and foster reproducibility in many areas of biomedical research.

The manuscript is well-written, and the figures are clear and easy to understand. Although I am excited about the results presented, I have some concerns that the authors should address before becoming convinced of AIMOS's novelty, strength and utility.

We are glad to hear that the reviewer is as excited about the results as we are. Moreover, we are grateful for the valuable comments, which helped us to further strengthen the manuscript, and hope that we could address any concerns.

Major comments

1. The authors compare the AIMOS method with many other deep-learning tools and they write that "learnable convolutional kernels provide the most promising basis for automatic organ segmentation". But it is not clear to me what new benefits and breakthroughs are offered by the AIMOS method. In fact, reference 42 shows that the W19 method has very similar performance as AIMOS. What are the main benefits of AIMOS? It seems that the training used for AIMOS is quite much greater than for W19 (TS-DSN) (140 vs. 40). Is it possible that the larger training performed by AIMOS is the reason why AIMOS performs better than TS-DSN? Indicate better benefits and breakthroughs offered by the AIMOS method.

AIMOS is optimized for usability, adoptability, and interpretability. It is the fastest and most accurate completely automated method to segment mouse organs reported so far. It matches or exceeds the segmentation quality of human experts, which has not been demonstrated for any other method before. Importantly, while we validate the performance on the largest dataset reported so far, we show that AIMOS can be trained from scratch efficiently even with very small datasets (8-12 scans) and that the pre-trained models, which we provide, reduce this need even further. At a dataset size of 40 scans (which was used in W19), AIMOS already reaches 100% of its final performance (see Supplementary Figure 1c).

However, the benefits of AIMOS go beyond "just" improving segmentation performance along two dimensions. First, it is a very versatile tool optimized for easy and wide-spread adoption: it works across imaging modalities, it can segment a very large number of 11 anatomical structures, it works for a broad range of resolutions (here tested for a range from 140 μ m/px to 1,120 μ m/px), it has very low computational requirements (it works on a single GPU; but it can be run even on a CPU), and is based on a simple-to-implement single stage network. By contrast, W19 requires 7 different networks and a server with 3 GPUs. Second, our study and AIMOS as a tool offer insight into the uncertainty of the organ segmentations caused by human error and bias due to ambiguity in the data. As we discuss in detail in the manuscript, prior approaches did not take this into account for a) model evaluation and b) downstream statistical analysis of the derived organ segmentations. This is an important and central contribution by this study. AIMOS as a tool takes this up by providing localized measures of this uncertainty, enabling to take it into account for future studies.

We thank the reviewer for this input and appreciate that these differences may not have been stated sufficiently clear in the manuscript before. We have improved the manuscript, especially in the introduction (paragraph 2) and discussion (paragraph 1) in order to make these benefits of AIMOS clearer.

2. I think there are some statements that are exaggerated and misleading that should be reworded. For example, line 235, "While AIMOS exceeds prior art and human performance by a margin ..." and line 18, "with unrivalled accuracy".

We agree that this wording is not ideal and have toned down/removed these statements.

3. The introductory section does not mention why this work is important and how the AIMOS method can contribute to science. What are the weaknesses and limitations of existing segmentation methods? How can AIMOS bridge this gap?

In short, AIMOS is important as it addresses the long-standing need for automated, high-quality segmentations of multiple organs in mouse scans. Existing methods do not reach satisfactory segmentation quality, often have substantially slower processing speed, and do not take into account intrinsic ambiguity in the data, which leads to diverging human interpretations. AIMOS bridges this gap by providing high-quality segmentations in less than 1 second and, importantly, yielding additional insight by localizing and quantifying uncertainty for these segmentations related to diverging human interpretation due to ambiguous data.

In the original manuscript, we had focused on highlighting the improvements and benefits of AIMOS rather than the shortcomings of prior methods. But we appreciate that such a comparison may be helpful for the reader and thank the reviewer for this comment. We have re-written parts of the introduction (paragraph 2) in order to show more explicitly which shortcomings of prior methods are overcome by AIMOS and how it contributes to science.

4. In this manuscript, the AIMOS segmentation method is challenged with micro-CT datasets. What should be emphasized is that micro-CT images have quite low contrast and are difficult to segment, even for human experts. But this fact is mentioned only for the spleen and that the human annotators had low segmentation performance for the spleen. The authors need to explain this better to sell their story to a wide readership. If information about the difficulties in segmenting micro-CT data sets, even for human experts, is mentioned, it strengthens the AIMOS method.

We thank the reviewer for raising this important point. We agree that (micro-)CT datasets are intrinsically hard to segment, even and especially for humans; CNN-based approaches, however, are well suited to deal with low-contrast CT data. We have adapted parts of the introduction (paragraph 2) and the discussion (paragraph 1) to make this point clearer.

5. The authors state that the AIMOS method is orders of magnitude faster than prior algorithms. Processing times should be compared to other methods that use deep learning, not just compare it with manual and atlas registration methods. It is already well known that methods based on machine learning are much faster than manual segmentation and the registration method.

We agree with the reviewer that machine learning methods in general tend to be faster than other segmentation methods and that this also applies to AIMOS. We compare against manual and atlas-registration-based methods as these are still by far the most commonly used approaches in labs for organ segmentation. With this, we want to motivate users to switch to faster machine learning based approaches. But of course we agree that other learning based approaches can be similarly fast (in the order of seconds instead of minutes or hours) and have thus adjusted the text (Results, Quantitative Segmentation Performance, paragraph 4) and Figure 2 in the manuscript accordingly.

6. Figure 6 shows a full-body scan of a mouse recorded with a light sheet microscope. The light-sheet microscopy images have higher contrast than the micro-CT images, and AIMOS can correctly segment organs from light-sheet data. However, the "human expert" appeared to have difficulty selecting bodies in the light-sheet images; the human expert chose the larger area with a black background. This can lead to poor scores for metastatic quantification of the human expert. Then it will be too much to say that AIMOS exceeds the segmentation quality for human experts. Did the human experts really do their best segmentation? Describe how the segmentation was performed.

The segmentation was carried out diligently and completely manually on a slice-by-slice basis (regularly spaced, interpolated in between, with subsequent refinement iterations). We realize that Figure 6 may lead to a different impression at first sight, which we aim to improve in the revised version of the manuscript.

The panels in Figure 6d show mean-intensity projections of the volumetric region around the organ. This kind of display is needed in order to display the positions of all metastases in and around that organ. Thus, the segmentations are also shown as projections. For the prediction, we show the projection of the volumetric segmentation as semi-transparent shades. To ensure good visibility of underlying imaging data, both segmentations, and the metastases, we chose to show the human annotation as the outline of its axial projection (solid line). Thus, it marks the outer-most extent of the organs from the axis of projection. In some cases, this may lead to the conclusion that the organ is over-segmented by the human (most prominently for the lower part of the left kidney). A slice-wise comparison shows that this is not the case (left: raw imaging data; middle: imaging data overlaid with human expert annotation; right: imaging data overlaid with AIMOS prediction):

This comparison shows that the outline is indeed fairly precise (for human annotation as well as for AIMOS prediction). The visual impression in Figure 6 (in which the lower part seems rather dark) is driven by the fact that the left kidney only extends this far for a few slices. In a mean-intensity projection, this may appear dark in some cases. We have experimented with alternative ways of presenting this data but found the current one the best.

We appreciate that the original manuscript presented this Figure in a way that may have led to wrong interpretation and thank the reviewer for pointing this out. We have now improved the revised version of the manuscript to address this more clearly. To this end, we have extended the description of the annotation process in detail. Also, we now describe the details of the visualization in Figure 6 in the Methods and we have adjusted the description of the Figure in the main text and the Figure legend accordingly.

Minor

Line 18, The phrase “unrivalled accuracy” in the abstract feels a too strong statement.

We have adjusted the wording.

Line 29, The start of paragraph two in the Introduction feels abrupt. Maybe the last sentence of paragraph one can go to the beginning of paragraph two.

We have improved the flow of the text as suggested.

Line 200 and Figure 1, change “fluorescent light-sheet microscopy” to “fluorescence light-sheet microscopy”

We have adjusted the text.

Figures 1 and 6d, Supplementary Figure S3f,g,h, add scale bars.

Figures 4a,b, 5b,c,d, Supplementary Figure S2, please add scale bars in all panels. Scale bars are present in all other panels, even though they have the same size as neighboring panels, e.g., Fig 3a and b.

We have adjusted the Figures accordingly and added all scale bars.

Line 219, change “...we the compared...” to “...we then compared...”.

We have adjusted the wording.

Figure 2a, 2b, 6b, S3a-d, S4a, the numbers on the y-axis are too small.

We have increased the font size of the labels on the y- and, where appropriate, on the x-axis.

Figure S3e-h, add a space before and after “=”.

We have added a space.

Figure S4b, “t” should be in italic.

It is now in italic.

Reviewer #2 (Remarks to the Author):

The authors present a software tool for fully automated mouse organ segmentation for whole-body scans, e.g. from μ CT. Such organ segmentation is frequently required in preclinical research and an accurate and robust automated method would save manual segmentation effort and remove a large potential source of bias because manual segmentation suffers from substantial inter-reader variability. The authors use a large freely available μ CT data base, recently published for this purpose, including scans with and without contrast agent. The authors claim superior segmentation performance compared to previous approaches. Furthermore, they show that annotations from more than one person should be used to better evaluate the performance of automated methods which is a relevant finding. The code is published to enable reproduction of the results which is appreciated. The study, including the software tool, is a highly relevant contribution to the applied imaging community but I recommend several improvements of the manuscript.

We are delighted to hear that the reviewer sees the study and AIMOS as a tool as a highly relevant contribution and we are thankful for the suggested improvements to further strengthen the manuscript.

In the abstract, it would be good to highlight the aspect that manual annotations from multiple humans should be used to evaluate automated segmentation methods.

We agree that this important finding of the study should be mentioned in the abstract. It is common practice in the field to only use a single human expert segmentation and thus, this finding should be communicated more prominently. We have adjusted the abstract accordingly.

In the introduction, manual segmentation by delineation in each slices is mentioned. In many applied studies, interactive segmentation is actually used which includes operations like region growing and cutting. This strongly reduces the time for irregularly shaped organs like lungs and bones which have a strong contrast to their surrounding. In that light, the mentioned duration of 9 hours for manual segmentation seems to be too long as assumption. In our studies we take around 30 minutes per scan using interactive segmentation. Nevertheless, an automated segmentation would be highly appreciated.

The reviewer is right in stating that interactive segmentation tools are often used and reduce the time needed to segment scans. Especially for (semi-)manual segmentation, there is an inherent trade-off between time and quality. Interactive segmentation with region growing was used, for instance, for the lungs in the public data set of Rosenhain et al (2018). This reduces the time needed for segmentation. However, it can also lead to sub-optimal segmentations – which was indeed observed for this dataset (see Figure S4b). We explicitly decided not to further refine this public dataset in order to maintain full reproducibility. Given the substantial time investment to create such large datasets, interactive segmentation is a very understandable and justified approach.

For the light-sheet microscopy data, however, we did not make use of interactive segmentation tools in order to avoid such problems. We want to emphasize two aspects. First, not every single slice was segmented as this would indeed take even longer; we segmented regularly spaced slices, interpolated in between, and subsequently refined the resulting segmentation. This process allowed a good trade-off between high quality and

acceptable segmentation times. Second, the 9 hours of total segmentation time per mouse for the light-sheet microscopy were not assumed but measured; this time reflect all steps as mentioned before.

We agree with the reviewer that the manuscript would benefit from describing the segmentation procedure as well as the time assessment in more detail. Before, it may have seemed like the 9 hour segmentation time was actually caused by segmenting every single slice independently. We have adjusted the manuscript accordingly, added this information (Methods, Data, paragraph 2), and also we now explicitly address that interactive segmentation tools are also commonly used and can reduce the time for (semi-)manual segmentation (Introduction, paragraph 2; Results, Quantitative Segmentation Performance, paragraph 4). We thank the reviewer for pointing this out.

Please clarify that the segmentation scores from other approaches are based on other data sets and therefore not entirely comparable because CT scans can have strongly varying quality. It would be highly interesting to see the performance of some of the other approaches on the same (new) data set.

It is true that the segmentation scores reported in prior literature are based on other datasets (which are not publicly available). We have made this clearer by mentioning it in the Figure legend and in the text (Results, Quantitative Segmentation Performance, paragraph 5). The comparison provided in Figure 2D is still a strong and in our eyes valid indication of model performances. Please note that these scores reflect the performance of models highly optimized on that specific datasets (consider hyper-parameter tuning) and may thus be seen as an upper limit for generalization on other datasets. For this reason, we not only benchmark our model on a public dataset but also make our own data available as we believe this can improve reproducibility, comparability, and thereby foster future development of such models. With their public CT dataset, Rosenhain et al. (2018) have laid the foundation for open comparison and we want to contribute to this by also making our data completely public.

In page 9, line 303, please clarify how the prediction is evaluated against both annotators. Is this done on a voxel-based decision?

Yes, this is done on a voxel-based decision. We have clarified this in the manuscript.

Results would be interesting for the urinary bladder which humans are quite good and fast at segmenting. Our own unpublished automated approaches failed miserably unfortunately for the bladder. The published data base also contains manual annotations of the bladder. In case the bladder turns out to be difficult for AIMOS, please discuss why this is the case and how it could be resolved in future work.

We have extended AIMOS to also segment the bladder. Without any further changes in model or setup, AIMOS can segment the bladder of the native CT scans at high quality with a Dice score of 89% - again, exceeding prior literature and human performance. A similarly good result was achieved for the contrast-enhanced CT dataset (88%). We have included these results into the manuscript (Figures 2, 3, 4, and 5, and corresponding text). This quick adaptation of AIMOS indicates its flexibility and generalizability for additional purposes.

Importantly, while the AIMOS prediction of the bladder works well and consistently, there is one case in which the AIMOS prediction does not overlap at all with the human annotation (Dice score of 0%; mouse #6 at t=072h). We have analyzed this case in detail and have come to the conclusion that this is caused by an error in the human annotation and that the AIMOS prediction actually captures the true location of the bladder. We reached out to the authors of Rosenhain et al. (2018) and they had kindly confirmed that the annotation was indeed not correct. This shows nicely how AIMOS can also be used to detect such outliers. We report this defective label in Supplementary Figure S5 (along with other cases where a low Dice score is explained by the mismatch of a defective label with a reasonable prediction). Please see below for details (left: raw imaging data; middle: imaging data overlaid with human annotation; right: imaging data overlaid with AIMOS prediction):

Please clarify that one dice score was computed for each organ, using only the segmentation of that organ.

Yes, the Dice scores were computed for each organ (and each scan) – using only the segmentation of that organ. We have made this explicit in the Methods section.

In the formulas for the variability, the time index should either start at 1 like the other indices or the superscript of the summation operator should be T-1 instead of T.

We agree with the reviewer and are thankful for pointing this out. We have adjusted the index.

For the contrast-enhanced scans, there seems to be a misunderstanding what pre-injection means. In the used data-based, pre-injection time point means that it was before fluorescent contrast agent injection, i.e. after μ CT contrast injection. That explains why the spleen is visible in the so-called pre-injection scans.

We thank the reviewer for pointing this out. We have re-confirmed the exact procedure with the authors of that study and refined the wording throughout the manuscript to avoid any potential misunderstandings.

I am not sure if bio-distribution is the right term for cancer metastases. I would rather call it organ-distribution. Bio-distribution is typically used for the distribution of an injected probe.

We have adjusted the wording as recommended and now refer to it as "organ distribution of metastases" or "spatial metastasis distribution"

Please discuss future work, e.g.

-Using a 3D network instead of 2D-networks per slice. This could help for complex 3D shapes where a single slice is providing insufficient information. And it could avoid flickering between slices.

-Could combination with an atlas-based method further improve the results?

We have added these suggestions to the Discussion (paragraph 3). Generally speaking, we believe that the 2D architecture is a better choice than a 3D architecture at this time as it can be trained with less data and has a substantially lower memory profile, enabling researchers to run AIMOS on a regular workstation. However, 3D architectures of course have some benefits and, importantly, 3D postprocessing of 2D predictions may provide options to combine the "best of both worlds". Similarly, a combination with atlases may also yield further potential. We now discuss this in the revised manuscript.

One important aspect of this discussion, though, is to properly define what "improved results" actually mean. As we stress in the manuscript, an even higher Dice score may only seemingly be better. With prediction scores as high as reported in this study, further increasing overlap to a given reference may just represent "overfitting" to that reference. In our eyes, avenues to further improve segmentation pipelines may include 3D architectures but especially approaches for stronger generalization and uncertainty analysis. We included these ideas as suggested in the manuscript to provide more inspiration for future work.

Reviewer #3 (Remarks to the Author):

I Summary

The manuscript employs a U-Net architecture to perform mouse whole-body segmentation from both micro-CT and light sheet microscopy volumetric images. The system is evaluated on a large set of human-annotated data in terms of DICE overlap. An additional "uncertainty" output channel is used to predict ambiguous regions.

II Overall Comments

Clarity

The manuscript is very well-written and structured and very simple to follow. The figures are clear and concise. The technical content is simple to grasp.

Originality

The U-Net architecture is well-established and heavily used for various segmentation tasks. The AIMOS system is a slight modification of this architecture. The additional uncertainty channel is an interesting and useful add-on. Overall, the contribution of the paper is the application of a known architecture to a dataset relevant for research.

Significance

The proposed system has the potential to become a standard tool for mice research and quantification due to its simplicity, accuracy and speed. However, there are more labels in the dataset(s) [36, 37]. The impact of the system could be significantly increased if further (more fine-grained) labels such as left/right eye, left/right kidney [36] and brain, tumor, stomach, vena cava, bladder [37] as well as further skeleton labels such as upper/lower limbs, ribs, sternum central [37] would be included. Furthermore, a more exhaustive analysis of the uncertainty prediction and different performance measures would help to better understand strength and weaknesses.

We are very excited to hear that the reviewer sees the potential of AIMOS becoming a standard tool for murine research. We aimed to explicitly design the entire pipeline around the principles of simplicity, ease-of-use, low computing requirements, and reproducibility in order to foster maximum adoption.

As suggested by the reviewer, we now include more (/more fine-grained) labels (left/right kidney, bladder, stomach, intestine) in the experiments shown in Figures 2, 3, 4, and 5. The studies [36, 37] indeed report further labels (eyes, vena cava, granular skeleton labels). Unfortunately, the authors of both studies have not made their data publicly available. Thus, it is not possible for us to make use of it. The public dataset by Rosenhain et al. (2018), which is part of the data we use to validate the AIMOS approach, does not contain any labels for eyes, brain, vena cava, or sub-divided skeleton labels.

In the original manuscript we had focused on the organs that are most commonly reported by other literature on segmentation methods and match the labels we had created for our own light-sheet microscopy dataset (used for tumor metastasis research). This is the reason why we had not included further anatomical structures so far. However, the reviewer makes an important suggestion that a large number of anatomical labels can further increase the utility

of AIMOS. To this end, we extended AIMOS to the maximum number of anatomical labels available in the given data:

- *Brain.* Please note that the scans of Rosenhain et al. (2018) does not include the head. However, AIMOS segments the brain successfully in the light-sheet fluorescence microscopy data, which we added and made publicly available.
- *Left/right kidney.* We have split the segmentation results for the left and the right kidney and now report both scores separately as requested. (left kidney: 89% / 90%; right kidney: 89% / 86%; for native / contrast-enhanced micro-CT)
- *Bladder.* We have extended AIMOS to also segment the bladder in both, native and contrast-enhanced micro-CT. Just as for the other organs, our segmentation performance (89% / 88%; for native / contrast-enhanced micro-CT) exceeds prior art (51-71%) and human segmentation performance (82% / 85%; for native / contrast-enhanced micro-CT).
- *Stomach.* We have extended AIMOS to also segment the stomach in contrast-enhanced micro-CT. Just as for the other organs, our segmentation performance (79%; for contrast-enhanced micro-CT) exceeds prior art (76%) and human segmentation performance (68%). Please note that the labels for native micro-CT are largely missing in Rosenhain et al. (2018) and therefore we restricted this analysis of the stomach segmentation to the contrast-enhanced micro-CT dataset.
- *Intestines.* In addition to reviewer's requests, we also worked on the intestines to maximize the amount of anatomical labels reported. While we cannot compare to prior art since none of the studies of our comparison in Fig 2d had segmented intestines, we can still show that our segmentation performance (88% / 87%; for native / contrast-enhanced micro-CT) exceeds human segmentation performance (72% / 68%; for native / contrast-enhanced micro-CT).
- *(Trachea.* In addition to reviewer's requests, we hoped to include the trachea. While Rosenhain et al. (2018) list the trachea as a labeled structure, unfortunately the labels were largely incomplete so that we decided not to include it in our study.)

We hope that the good segmentation results we now added for left/right kidney, bladder, stomach, and intestines further increase the reviewer's confidence that AIMOS generalizes well across various organs. We agree with the reviewer that the increased number of anatomical labels make the manuscript stronger and are thankful for this suggestion.

We also now include a more exhaustive analysis of the uncertainty prediction and report further performance measures as requested – please refer to the sections "Empirical analysis", parts a) and c), of the response to this review further below for a detailed response.

[36] Van Der Heyden, B. et al. Automatic multiatlas based organ at risk segmentation in mice. *The Br. journal radiology* 92, 20180364 (2018).

[37] Akselrod-Ballin, A. et al. Multimodal correlative preclinical whole body imaging and segmentation. *Sci. reports* 6, 27940 (2016).

Reproducibility

The data is public and the code is shared on github. The capsule on CodeOcean allows to interactively run the model. So the work can be considered fully reproducible.

We are glad to hear the reviewer considers the work fully reproducible. We have designed the entire study around this goal and thus also stick to published data (Rosenhain et al. (2018)) and make our own data public.

Empirical analysis

The manuscript provides an analysis of the segmentation accuracy in terms of DICE score with respect to two human annotators and a comparison to 8 different segmentation algorithms on micro-CT data. Furthermore, an assessment of the validity of the ambiguity prediction is provided and results for light-sheet microscopy are discussed.

a) While the analysis is concise and exhaustive, some aspects could be improved. There are accuracy measures complementing the DICE score e.g. [36] reports Hausdorff distance percentiles and center of mass displacement. An additional analysis with different performance metrics would make the paper much stronger.

As suggested by the reviewer, we now have included the Hausdorff distance and the center of mass displacement as additional metrics. For this, we added a new Supplementary Figure (S4) to the manuscript. In general, we observe that the Hausdorff distance and the center of mass displacement are mostly well below 1mm and therefore indicate a high quality of predictions – in line with the high Dice scores.

For the Hausdorff distance it is especially important to mention that it may be very sensitive to some of the defective labels we have seen (and also reported). The Hausdorff distance is a measure for the distance between the surfaces of the prediction and the reference annotation. However, in some cases the annotations are defective in such a way that a surface-based measure may be drastically affected. For example, as also reported in the original submission, this occurs in the lungs (see Supplementary Figure S5b). But even beyond that, we sometimes observed erroneous annotations with rather distant surfaces, for example in the kidneys of mouse #8 at t=168h in the contrast-enhanced dataset (maximum-intensity projection of binary mask for kidneys; left: human annotation; right: AIMOS prediction):

Given that the debris-like pattern in the annotation has a rather large surface compared to the solid areas, this defect in the label may heavily affect the Hausdorff distance (even when reporting the 95th percentile). We thus also report the 50th percentile, which quantifies the median distance of surfaces between prediction and annotation and is thus less sensitive to such outliers. The median distance between the contours is around 0.0-0.3mm, which is in

the range of the size of 1-2 voxels (voxel size is 140 μ m for the CT data). This means that the surfaces of prediction and annotation at a representative location are only a 1-2 pixels apart.

b) An ablation study for the slice-wise preprocessing with intensity normalization and resampling should be provided to establish that this is a required step.

As the reviewer states, the preprocessing module consists of two steps: signal intensity normalization and resampling.

The first step, intensity normalization, is a common pre-processing procedure for deep neural networks. With a standard z-score normalisation, the input for AIMOS was normalised to zero mean and unit standard deviation. This has been proven to facilitate efficient training in previous studies (see [1-4]), and it is also in line with the batch normalization that is also used (see [5]).

The second step, resampling, is a data-specific procedure and should indeed be analyzed in detail. Resampling can be effectively seen as a trade-off between level of detail and model complexity. Providing data at original resolution enables to preserve a maximum of information for the model. Reducing resolution can potentially remove information that may be helpful for the organ segmentation task. However, there is also a desire to keep model complexity low. Larger images in higher resolution require more computation complexity to be processed effectively (considering the effective field of view of the inner-most convolutional kernels). Thus, keeping scans at their original (high) resolution may not be the most efficient representation of the information required to solve the task. The optimum of this trade-off cannot be trivially predicted and thus, the reviewer is right that an ablation study regarding the resampling can be insightful.

We had provided such an analysis in Supplementary Figure S1a. We have adjusted the manuscript to shed more light on this trade-off and how we analyzed it (Results, paragraph 1). In short, we show that AIMOS works reliably over a large range of resolutions (down to the range of 1mm³/voxel) – giving users the possibility to process scans of various original resolutions and also giving them the possibility to resample the scans to a resolution of their choice. In that sense, the resampling step is not required per se in order to use AIMOS but is rather a feature that allows to process scans of various resolutions (as in our case, ranging from (140 μ m)³/voxel to (1,120 μ m)³/voxel). A further ablation analysis, focusing on network complexity (the second part of the mentioned trade-off) is provided in Supplementary Figure S1b.

*[1] Jayalakshmi, T., and A. Santhakumaran. "Statistical normalization and back propagation for classification." *International Journal of Computer Theory and Engineering* 3.1 (2011): 1793-8201.*

*[2] LeCun, Y., Bottou, L., Orr, G., and Muller, K. Efficient backprop. In Orr, G. and K., Muller (eds.), *Neural Networks: Tricks of the trade*. Springer, 1998b.*

*[3] Wiesler, Simon and Ney, Hermann. A convergence analysis of log-linear training. In Shawe-Taylor, J., Zemel, R.S., Bartlett, P., Pereira, F.C.N., and Weinberger, K.Q. (eds.), *Advances in Neural Information Processing Systems 24*, pp. 657–665, Granada, Spain, December 2011.*

[4] Falk, T., Mai, D., Bensch, R., Çiçek, Ö., Abdulkadir, A., Marrakchi, Y., Böhm, A., Deubner, J., Jäckel, Z., Seiwald, K. and Dovzhenko, A., 2019. U-Net: deep learning for cell counting, detection, and morphometry. *Nature methods*, 16(1), pp.67-70.

[5] Ioffe, Sergey, and Christian Szegedy. "Batch normalization: Accelerating deep network training by reducing internal covariate shift." *arXiv preprint arXiv:1502.03167* (2015).

c) Similarly, the effect of the ensembling and median smoothing over ten models should be properly analyzed. How much does the performance depend on this? How strongly is the runtime affected?

As stated in the manuscript, the ensemble-voting procedure is an optional feature; we want to highlight here that it is not needed per se to derive high quality organ segmentations. It serves to even out rare outlier predictions. In short, we consider this feature as optional as it only has a small effect on the reported median performances (around 1-2 p.p.; see the newly added Supplementary Figure 4). This improvement largely comes from evening out rare outlier predictions – which may be a useful feature for some users of the technology in which negative outliers would limit the utility of automated methods. The effect on the runtime depends on the specific implementation but can generally be considered a) sub-linear, as the runtime is driven by general processing (which stays constant) plus network inference (which can multiply with ensemble size if not executed in parallel) and b) negligible from a practical perspective since AIMOS segments a mouse body in the order of seconds (rather than minutes as with non-CNN based methods). We appreciate the reviewer's suggestion that the effect on performance should be reported more explicitly and have added such an analysis to the manuscript text. To this end, we now also report the effect of ensemble-voting for each organ and each dataset in a newly introduced Supplementary Figure (S4). Moreover, we reference further literature that explicitly focusses on the details of ensemble approaches in biomedical image segmentation.

d) The uncertainty prediction should be analyzed in more depth to establish its value and limits. It would be great to quantify with numbers whether the ambiguity information can be used in practice beyond visual plausibility. For example, how much of the discrepancy between human annotators and prediction falls inside the ambiguity region. Are all voxels which the network classifies as being non ambiguous correctly predicted?

We appreciate the ask for quantitative analysis on the uncertainty prediction and its value for applications. Indeed we believe that it can be used beyond visual plausibility (even though this surely is an important benefit). A central use case, organ volumetry, is already provided in Figure 5f. Organ volumetry is often used to assess pathological developments (e.g., see [1]). Here, we show that the ambiguity may cause differences in estimated organ volume that are bigger than natural variability across animals. Thus, the AIMOS prediction of ambiguity-driven uncertainty can be used to refine statistical analysis in order to assess whether observed differences between a control and a test group are actually significant given the uncertainty in the segmentations (which is, to the best of our knowledge, currently not a common practice). We have refined the text in the manuscript to make this use of AIMOS uncertainty prediction clearer (Results, newly introduced subsection "Quantitative application of AIMOS for uncertainty quantification").

Of course we fully agree with the reviewer that this is only of use if the predicted ambiguous regions actually match the discrepancy of human annotators. We can assess the binary

overlap of the thresholded uncertainty prediction with the actual disagreement of both annotators (Dice score). If we set the threshold at 50%, the volumetric overlap amounts to a Dice score of 55% for contrast-enhanced and 60% for native micro-CT. Given that ambiguous regions have per definitionem no clear delineation, this Dice score is a very good indicator of the usefulness of the uncertainty prediction – and in line with the observed usefulness given the qualitative comparison (Fig 5c,d). We have added this quantitative assessment along with the limitations to the manuscript (Results, AIMOS predicts the regions of diverging human interpretation, paragraph 1) and thank the reviewer for this suggestion.

The question, whether all voxels in regions classified as non-ambiguous, are predicted correctly, can only be answered with some limitations. Generally, we argue that this is not the case for **all** voxels, as errors occur in any method – including AIMOS. But there are two aspects to consider here: first, the question can only be answered with limitations due to the very reason that it is not clear what the "correct" prediction would actually be – even if two independent annotators agree (since a third one might disagree). In that sense, a core message of our study is the non-existence of an absolute "ground truth". However, and this is the second consideration, there are surely regions in the scans that will be considered undisputed from any sufficiently trained human expert (say, the center region of a clearly visible organ). It is impossible to locate the exact border between undisputed and ambiguous regions as they can be considered a continuum. But the aforementioned Dice scores and the comparison of panels b), c) and d) in Figure 5 clearly suggests that the residual error of the AIMOS organ prediction spatially correlates with ambiguity-driven mismatch of human annotations around the borders, which again is well predicted. Thus, while surely not **all** voxels classified as non-ambiguous are predicted correctly, a very high share of them is and the residual error is mostly driven by voxels classified as ambiguous.

We hope that this answers the reviewer's question in a satisfactory manner and that the newly added quantitative analysis of predicted human disagreement meets the reviewer's expectation.

[1] Masi, Brice, et al. "In vivo MRI assessment of hepatic and splenic disease in a murine model of schistosomiasis." *PLoS Negl Trop Dis* 9.9 (2015): e0004036.

Quality

The manuscript presents an interesting contribution but would benefit from some more empirical assessment.

We are glad to hear the reviewer appreciates the contribution of this study and have added the suggested additional analyses. We thank the reviewer for these suggestions and hope that the revised manuscript meets the reviewer's expectations.

III Detailed Comments

- 1) The authors report that the AIMOS system is faster and more accurate than previous systems but the precise analysis why the prediction is faster is not given. Is it simply the use of a standard U-Net that is so fast or are there additional tweaks required?

Convolutional neural network such as the U-Net are known as a very fast way of analyzing imaging data (e.g., see [1]). One benefit provided by our approach is to enable the

exploitation of a U-Net-like architecture for the task of multi-organ segmentation. AIMOS is so fast as it is the first end-to-end single stage deep learning solution to the problem, which does neither require multiple stages nor computationally expensive pre- or postprocessing (e.g., registration, transforms, cascaded networks). In that sense, there are no "tweaks" needed to achieve this speed but the "tweaks" presented in the paper enable the use of a single, end-to-end network, which is inherently fast. We have revised the manuscript text to make this more clear (Results, Quantitative Segmentation Performance, paragraph 4).

[1] Canziani, A., Paszke, A. and Culurciello, E., 2016. An analysis of deep neural network models for practical applications. arXiv preprint arXiv:1605.07678.

- 2) The term "uncertainty quantification" in the title is a bit of a bold claim. The paper does not quantify uncertainty in a proper probabilistic framework, the paper only qualitatively reports that the predicted ambiguity regions are reasonable. The term "quantification" would require a much deeper analysis. A better term would be something along the lines of "ambiguity forecasting" or "sensitivity estimation" instead.

We have removed the term "uncertainty quantification" from the title. While AIMOS indeed localizes and quantifies uncertainty (e.g., see the application for organ volumetry in Figure 5f), the reviewer is right that this differs from probabilistic approaches to uncertainty analysis. This difference may require explanation and thus, we have removed this from the title (where such explanation would exceed any sensible length of the title) in order to avoid any misunderstandings. In the further manuscript, we have added more detailed explanations to further clarify the terms used.

- 3) Is there a special region, why only 6 labels (heart, lungs, liver, kidneys, spleen, bones) were used? Shouldn't the use of more detailed label information allow to boost the segmentation performance of lumped labels such as "bones" overall?

As described in more detail in the upper section of this response letter (Reviewer 3, section II "Overall Comments", part "Significance"), in the original manuscript we had decided to focus on the organs that a) are most commonly used in biomedical research, b) most commonly reported by other literature on segmentation methods, and c) match the labels we had created for our own LSFM dataset (used for tumor metastasis research). We have now extended the scope by an additional 3 labels (bladder, stomach, intestines). We want to highlight that this represents the maximum amount of labels available for this task in the dataset used and also that there is no breakdown of the labels for "bones" available.

Generally speaking, adding more labels may indeed help the network to locate and segment structures of interest. This would be the case, for example, if a given structure alone is hard to locate in a scan but easier to locate if a neighboring structure has already been identified successfully. In our study, we would not expect such cases since all structures are already identified with high confidence. Conversely, predicting more labels can also be considered a harder task for a network to learn – especially when the task complexity were to exceed the model capacity. If that were the case, adding more labels may actually be detrimental to the performance. Here, we would not necessarily expect such a dynamic as we had shown in Supplementary Figure S1b) that AIMOS capacity is more than sufficient for solving the task.

For example, in the revision of this manuscript, we extended the scope of the experiment on contrast-enhanced micro-CT from 6 to 10 anatomical labels. As expected, this did not

change the performance for the 6 labels in a meaningful manner – even though the anatomical structures are in direct proximity. While the performance for the kidneys decreased by 1p.p from 88% to 87%, it increased for the heart by 1p.p. from 91% to 92% and for the spleen also by 1p.p. from 88% to 89%. All other values did not change. We consider these changes to be of minor significance. To the contrary, these results indicate that the performance of AIMOS generalizes well across different anatomical structures and is not critically dependent on a given set of structures chosen for training and prediction.

[36] Van Der Heyden, B. et al. Automatic multiatlas based organ at risk segmentation in mice. *The Br. journal radiology* 92, 20180364 (2018).

[37] Akselrod-Ballin, A. et al. Multimodal correlative preclinical whole body imaging and segmentation. *Sci. reports* 6, 27940 (2016).

Reviewers' Comments:

Reviewer #1:

Remarks to the Author:

The authors have satisfactorily responded to all my comments and revised the manuscript accordingly

Reviewer #2:

Remarks to the Author:

The authors substantially improved the manuscript according the reviewers comments.

Minor points (typos only):

Page 2: "Here, we present a fully integrated pipeline base on a" should be "..based..."

Page 3: a dot is missing in "...study (see Results) During training..."

Reviewer #3:

Remarks to the Author:

The authors made a very strong effort to address all my comments and concerns. From my perspective, the manuscript is ready to get published given the typos are fixed.

- line 44: However, so far, there
- line 46: based on a single
- line 48: organs (brain, lungs
- line 129: (see Results).
- line 134: Why is 'Results' set in italics and not in line 129 above?
- line 408: Is the statement about CT or micro-CT.

REVIEWER COMMENTS – Response letter

We thank the reviewers for their positive feedback and for their final suggestions, which we all implemented.

Reviewer #1 (Remarks to the Author):

The authors have satisfactorily responded to all my comments and revised the manuscript accordingly.

Thank you for your kind support.

Reviewer #2 (Remarks to the Author):

The authors substantially improved the manuscript according the reviewers comments.

Minor points (typos only):

Page 2: "Here, we present a fully integrated pipeline base on a" should be "..based..."

Page 3: a dot is missing in "...study (see Results) During training...".

We have fixed these typos. Thank you for your kind support.

Reviewer #3 (Remarks to the Author):

The authors made a very strong effort to address all my comments and concerns. From my perspective, the manuscript is ready to get published given the typos are fixed.

- line 44: However, so far, there
- line 46: based on a single
- line 48: organs (brain, lungs
- line 129: (see Results).
- line 134: Why is 'Results' set in italics and not in line 129 above?
- line 408: Is the statement about CT or micro-CT.

We have fixed these typos. The newly revised version does not contain any italic words (in accordance with general formatting requirements). The statement in line 408 refers to micro-CT, which is now explicitly mentioned. Thank you for your kind support.